# SMPLer-X: Scaling Up Expressive Human Pose and Shape Estimation

**Zhongang Cai**[*,1,2,3], **Wanqi Yin**[*,2,4], **Ailing Zeng**[5], **Chen Wei**[2], **Qingping Sun**[2],
**Yanjun Wang**[2], **Hui En Pang**[1,2], **Haiyi Mei**[2], **Mingyuan Zhang**[1],
**Lei Zhang**[5], **Chen Change Loy**[1], **Lei Yang**[†,2,3], **Ziwei Liu**[†,1]
[1] S-Lab, Nanyang Technological University, [2] SenseTime Research, [3] Shanghai AI Laboratory,
[4] The University of Tokyo, [5] International Digital Economy Academy (IDEA)

## Abstract

Expressive human pose and shape estimation (EHPS) unifies body, hands, and face motion capture with numerous applications. Despite encouraging progress, current state-of-the-art methods still depend largely on a confined set of training datasets. In this work, we investigate scaling up EHPS towards the first *generalist* foundation model (dubbed **SMPLer-X**), with up to ViT-Huge as the backbone and training with up to 4.5M instances from diverse data sources. With big data and the large model, SMPLer-X exhibits strong performance across diverse test benchmarks and excellent transferability to even unseen environments. *1) For the data scaling*, we perform a systematic investigation on 32 EHPS datasets, including a wide range of scenarios that a model trained on any single dataset cannot handle. More importantly, capitalizing on insights obtained from the extensive benchmarking process, we optimize our training scheme and select datasets that lead to a significant leap in EHPS capabilities. *2) For the model scaling,* we take advantage of vision transformers to study the scaling law of model sizes in EHPS. Moreover, our finetuning strategy turn SMPLer-X into *specialist* models, allowing them to achieve further performance boosts. Notably, our foundation model SMPLer-X consistently delivers state-of-the-art results on seven benchmarks such as AGORA (107.2 *mm* NMVE), UBody (57.4 *mm* PVE), EgoBody (63.6 *mm* PVE), and EHF (62.3 *mm* PVE without finetuning). [2]

## 1 Introduction

The recent progress in expressive human pose and shape estimation (EHPS) from monocular images or videos offers transformative applications for the animation, gaming, and fashion industries. This task typically employs parametric human models (*e.g.*, SMPL-X [49]) to adeptly represent the highly complicated human body, face, and hands. In recent years, a large number of diverse datasets have entered the field [4, 6, 6, 61, 66, 37, 3, 12, 14, 14, 62, 7], providing the community new opportunities to study various aspects such as capture environment, pose distribution, body visibility, and camera views. Yet, the state-of-the-art methods remain tethered to a limited selection of these datasets, creating a bottleneck in performance across varied scenarios and hindering the ability to generalize to unseen situations.

Our mission in this study is to explore existing data resources comprehensively, providing key insights crucial for establishing robust, universally applicable models for EHPS. Accordingly, we establish the first systematic benchmark for EHPS, utilizing 32 datasets and evaluating their performance

---

[*]Equal contributions. [†]Co-corresponding authors.
[2]Homepage: `https://caizhongang.github.io/projects/SMPLer-X/`.

across five major benchmarks. We find that there are significant inconsistencies among benchmarks, revealing the overall complicated landscape of EHPS, and calling for data scaling to combat the domain gaps between scenarios. This detailed examination emphasizes the need to reassess the utilization of available datasets for EHPS, advocating for a shift towards more competitive alternatives that offer superior generalization capabilities, and highlights the importance of harnessing a large number of datasets to capitalize on their complementary nature.

Moreover, we systematically investigate the contributing factors that determine the transferability of these datasets. Our investigation yields useful tips for future dataset collection: 1) the more is not necessarily, the merrier: datasets do not have to be very large to be useful as long as they exceed approximately 100K instances based on our observation. 2) Varying indoor scenes is a good alternative if an in-the-wild (including outdoor) collection is not viable. 3) synthetic datasets, despite having traceable domain gaps, are becoming increasingly potent to a surprising extent. 4) Pseudo-SMPL-X labels are useful when ground truth SMPL-X annotations are unavailable.

Equipped with the knowledge procured from the benchmark, we exhibit the strength of massive data with SMPLer-X, a *generalist* foundation model that is trained using a diverse range of datasets and achieves exceptionally balanced results across various scenarios. To decouple from algorithmic research works, we design SMPLer-X with a minimalist mindset: SMPLer-X has a very simple architecture with only the most essential components for EHPS. We hope SMPLer-X could facilitate massive data and parameter scaling and serve as a baseline for future explorations in the field instead of a stringent investigation into the algorithmic aspect. Experiments with various data combinations and model sizes lead us to a well-rounded model that excels across all benchmarks that contests the community norm of limited-dataset training. Specifically, our foundation models demonstrate significant performance boost through both data scaling and model size scaling, reducing the mean primary errors on five major benchmarks (AGORA [48], UBody [37], EgoBody [66], 3DPW [56], and EHF [49]) from over 110 mm to below 70 mm (demonstrated in Fig. 1), and showcases impressive generalization capabilities by effectively transferring to new scenarios, such as DNA-Rendering [7] and ARCTIC [12].

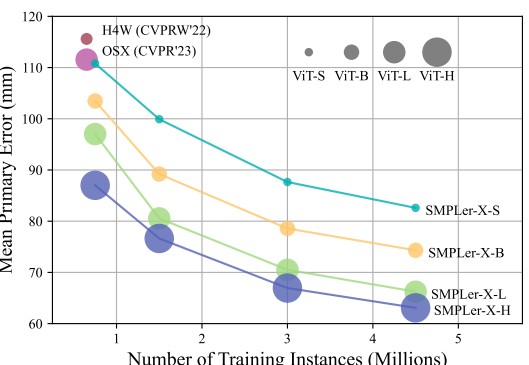

Figure 1: **Scaling up EHPS.** Both data and model scaling are effective in reducing mean errors on primary metrics across key benchmarks: AGORA [48], UBody [37], EgoBody [66], 3DPW [56] and EHF [49]. OSX [37] and H4W [44] are SOTA methods. Area of the circle indicates model size, with ViT variants as the reference (top right).

Furthermore, we validate the efficacy of finetuning our *generalist* foundation models to evolve into domain-specific *specialists*, delivering outstanding performance on all benchmarks. Specifically, we follow the same data selection strategy that empowers our specialist models to set new records on the AGORA leaderboard by being the first model to hit 107.2mm in NMVE (an 11.0% improvement) and achieving SOTA performance on EgoBody, UBody, and EHF.

Our contributions are three-fold. **1)** We build the first systematic and comprehensive benchmark on EHPS datasets, which provides critical guidance for scaling up the training data toward robust and transferable EHPS. **2)** We explore both data and model scaling in building the *generalist* foundation model that delivers balanced results across various scenarios and extends successfully to unseen datasets. **3)** We extend the data selection strategy to finetune the foundation model into potent *specialists*, catering to various benchmark scenarios.

## 2  Related Work

**Expressive Human Pose and Shape Estimation (EHPS).** Due to the erupting 3D virtual human research applications [64, 65, 19, 18, 5] and the parametric models (e.g., SMPL [40] and SMPL-X [49]), capturing the human pose and shape (HPS) [26, 31, 28, 29, 36, 57, 58], and additionally hands and face (EHPS) [49, 59, 8, 51, 68, 13, 54, 63] from images and videos have attracted increasing

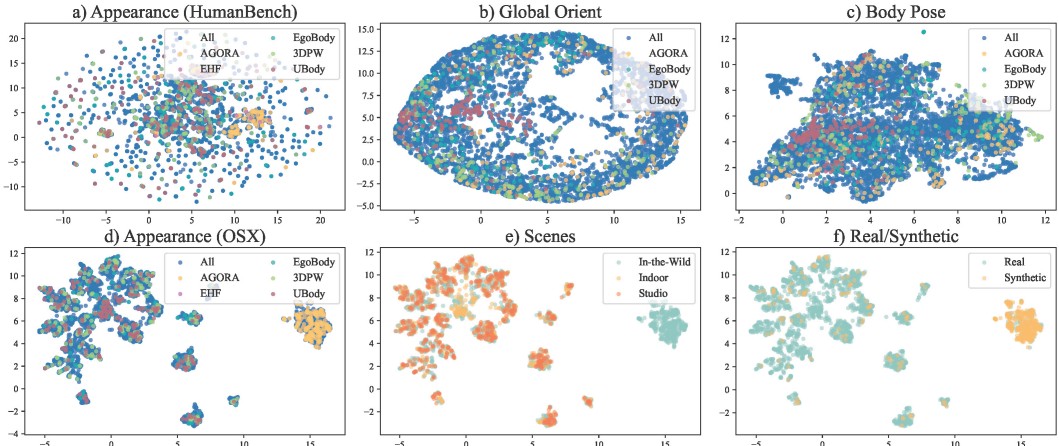

Figure 2: **Dataset attribute distributions.** a) and d) are image feature extracted by HumanBench [55] and OSX [37] pretrained ViT-L backbone. b) Global orientation (represented by rotation matrix) distribution. c) Body pose (represented by 3D skeleton joints) distribution. Both e) scenes and f) Real/Synthetic are drawn on the same distribution as d). All: all datasets. UMAP [41] dimension reduction is used with the x and y-axis as the dimensions of the embedded space (no unit).

attention. Optimization-based methods (e.g., SMPLify-X [49]) detect 2D features corresponding to the whole body and fit the SMPL-X model. However, they suffer from slow speed and are ultimately limited by the quality of the 2D keypoint detectors. Hence, learning-based models are proposed. One of the key challenges of EHPS is the low resolution of hands and face compared with the body-only estimation, making the articulated hand pose estimation and high-quality expression capture difficult. Accordingly, mainstream whole-body models first detect and crop the hands and face image patches, then resize them to higher resolutions and feed them into specific hand and face networks to estimate the corresponding parameters [8, 51, 68, 13, 54, 44, 63, 33]. Due to the highly complex multi-stage pipelines, they inevitably cause inconsistent and unnatural articulation of the mesh and implausible 3D wrist rotations, especially in occluded, truncated, and blurry scenes. Recently, OSX [37] proposes the first one-stage framework based on ViT-based backbone [11] to relieve the issues in previous multi-stage pipelines. This method provides a promising and concise way to scale up the model. However, they only use confined training datasets for a fair comparison and do not explore the combination of more data toward generalizable and precise EHPS.

**Multi-dataset Training for Human-centric Vision.** Recent efforts have been using multiple datasets in pretraining a general model for a wide range of downstream human-centric tasks. For example, HumanBench [55] leverages 37 datasets, whereas UniHCP [9] utilizes 33 datasets for tasks such as ReID, pedestrian detection, and 2D pose estimation. However, these works have only evaluated the efficacy of 2D tasks. Sárándi *et al.* [52] take advantage of 28 datasets in training a strong model for 3D keypoint detection, which recovers only the skeleton of subjects without estimating body shapes and meshes. Pang *et al.* [47] analyze 31 datasets for human pose and shape estimation (*i.e.*, SMPL estimation). However, hands and face estimation is not included, and only fewer than ten datasets are used concurrently in the most diverse training. This paper targets to scale training data and model size for EHPS, that simultaneously recovers the expressive pose and shape of the human body, hands, and face.

## 3 Benchmarking EHPS Datasets

### 3.1 Preliminaries

**SMPL-X.** We study expressive human pose and shape estimation via 3D parametric human model SMPL-X [49], which models the human body, hands, and face geometries with parameters. Specifically, our goal is to estimate pose parameters $\theta \in \mathbb{R}^{55 \times 3}$ that include body, hands, eyes, and jaw poses; joint body, hands and face shape $\beta \in \mathbb{R}^{10}$, and facial expression $\psi \in \mathbb{R}^{10}$. The joint regressor $\mathcal{J}$ is used to obtain 3D keypoints from parameters via $R_\theta(\mathcal{J}(\beta))$ where $R_\theta$ is a transformation function along the kinematic tree.

Table 1: **Benchmarking EHPS datasets.** For each dataset, we train a model on its training set and evaluate its performance on the *val* set of AGORA and *testing* sets of UBody, EgoBody (EgoSet), 3DPW, and EHF. Datasets are then ranked by mean primary error (MPE). Top-1 values are bolded, and the rest of Top-5 are underlined. #Inst.: number of instances used in training. ITW: in-the-wild. EFT [25], NeuralAnnot (NeA) [45] and UP3D [32] produce pseudo labels.

| Dataset | #Inst. | Scene | Real/Synthetic | SMPL | SMPL-X | AGORA [48] PVE↓ | ★ | UBody [37] PVE↓ | ★ | EgoBody [66] PVE↓ | ★ | 3DPW [56] MPJPE↓ | ★ | EHF [49] PVE↓ | ★ | MPE↓ |
|---|---|---|---|---|---|---|---|---|---|---|---|---|---|---|---|---|
| BEDLAM [4] | 951.1K | ITW | Syn | - | Yes | 164.7 | 4 | 132.5 | 8 | 109.1 | 2 | **98.1** | 1 | **81.1** | 1 | **117.1** |
| SynBody [61] | 633.5K | ITW | Syn | - | Yes | 166.7 | 5 | 144.6 | 11 | 136.6 | 4 | 106.5 | 5 | 112.9 | 5 | 133.5 |
| InstaVariety [27] | 2184.8K | ITW | Real | NeA | - | 195.0 | 9 | 125.4 | 4 | 140.1 | 9 | 100.6 | 3 | 110.8 | 4 | 134.3 |
| GTA-Human II [6] | 1802.2K | ITW | Syn | - | Yes | 161.9 | 3 | 143.7 | 10 | 139.2 | 8 | 103.4 | 4 | 126.0 | 12 | 134.8 |
| MSCOCO [38] | 149.8K | ITW | Real | EFT | NeA | 191.6 | 8 | 107.2 | 2 | 139.0 | 7 | 121.2 | 10 | 116.3 | 7 | 135.0 |
| EgoBody-MVSet [66] | 845.9K | Indoor | Real | Yes | Yes | 190.9 | 7 | 191.4 | 18 | 127.0 | 3 | 99.2 | 2 | 101.8 | 2 | 142.1 |
| AGORA [48] | 106.7K | ITW | Syn | Yes | Yes | **124.8** | 1 | 128.4 | 6 | 138.4 | 6 | 131.1 | 12 | 164.6 | 24 | 145.4 |
| Egobody-EgoSet [66] | 90.1K | Indoor | Real | Yes | Yes | 207.1 | 15 | 126.8 | 5 | **103.1** | 1 | 134.4 | 18 | 121.4 | 10 | 147.5 |
| RICH [20] | 243.4K | ITW | Real | - | Yes | 195.6 | 10 | 168.1 | 15 | 137.9 | 5 | 115.5 | 8 | 127.5 | 13 | 148.9 |
| MPII [2] | 28.9K | ITW | Real | EFT | NeA | 202.1 | 11 | 123.9 | 3 | 155.5 | 15 | 131.9 | 14 | 140.8 | 16 | 150.8 |
| MuCo-3DHP [43] | 465.3K | ITW | Real | Yes | - | 187.7 | 6 | 185.4 | 17 | 146.4 | 12 | 119.4 | 9 | 134.7 | 15 | 154.7 |
| PROX [17] | 88.5K | Indoor | Real | - | Yes | 204.1 | 13 | 180.3 | 16 | 151.8 | 13 | 132.5 | 17 | 122.3 | 11 | 158.2 |
| UBody [37] | 683.3K | ITW | Real | - | Yes | 207.0 | 14 | **78.7** | 1 | 145.6 | 11 | 149.4 | 23 | 132.1 | 14 | 158.5 |
| SPEC [30] | 72.0K | ITW | Syn | Yes | - | 161.5 | 2 | 146.1 | 12 | 154.8 | 14 | 139.7 | 21 | 197.8 | 27 | 160.0 |
| CrowdPose [34] | 28.5K | ITW | Real | NeA | - | 207.1 | 16 | 129.8 | 7 | 156.9 | 16 | 156.3 | 25 | 154.5 | 22 | 160.9 |
| MPI-INF-3DHP [42] | 939.8K | ITW | Real | NeA | NeA | 221.5 | 20 | 166.7 | 14 | 142.7 | 10 | 131.6 | 13 | 155.5 | 23 | 163.6 |
| HumanSC3D [15] | 288.4K | Studio | Real | - | Yes | 215.2 | 18 | 237.8 | 22 | 167.3 | 17 | 113.0 | 7 | 107.1 | 3 | 168.1 |
| PoseTrack [1] | 28.5K | ITW | Real | EFT | - | 218.1 | 19 | 161.0 | 13 | 180.8 | 21 | 150.2 | 24 | 149.9 | 21 | 172.0 |
| BEHAVE [3] | 44.4K | Indoor | Real | Yes | - | 208.3 | 17 | 205.8 | 20 | 175.8 | 19 | 132.0 | 15 | 145.0 | 18 | 173.4 |
| CHI3D [14] | 252.4K | Studio | Real | - | Yes | 203.3 | 12 | 264.7 | 25 | 175.7 | 18 | 122.6 | 11 | 121.0 | 9 | 177.5 |
| Human3.6M [21] | 312.2K | Studio | Real | Yes | NeA | 226.0 | 21 | 276.1 | 26 | 200.6 | 24 | 112.3 | 6 | 120.8 | 8 | 187.2 |
| DNA-R-HiRes [7] | 998.1K | Studio | Real | - | Yes | 230.0 | 22 | 278.2 | 27 | 179.2 | 20 | 134.5 | 19 | 149.7 | 20 | 194.3 |
| 3DPW [56] | 22.7K | ITW | Real | Yes | NeA | 234.0 | 23 | 259.3 | 23 | 192.6 | 23 | 140.6 | 22 | 142.9 | 17 | 207.2 |
| ARCTIC [12] | 1539.1K | Studio | Real | - | Yes | 308.5 | 29 | 200.7 | 19 | 186.4 | 22 | 202.5 | 26 | 182.5 | 25 | 216.1 |
| DNA-R [7] | 3992.0K | Studio | Real | - | Yes | 274.7 | 26 | 341.5 | 30 | 214.4 | 27 | 138.4 | 20 | 115.5 | 6 | 216.9 |
| UP3D [32] | 7.1K | ITW | Real | UP3D | - | 257.5 | 24 | 224.1 | 21 | 216.6 | 28 | 211.5 | 27 | 194.8 | 26 | 220.9 |
| Talkshow [62] | 3326.9K | Indoor | Real | - | Yes | 286.4 | 27 | 133.2 | 9 | 203.6 | 25 | 291.3 | 29 | 201.9 | 28 | 223.3 |
| FIT3D [16] | 1779.3K | Studio | Real | - | Yes | 329.7 | 30 | 404.0 | 31 | 213.8 | 26 | 132.1 | 16 | 148.1 | 19 | 245.5 |
| MTP [46] | 3.2K | ITW | Real | Yes | Yes | 272.7 | 25 | 284.9 | 28 | 273.2 | 29 | 265.2 | 28 | 244.6 | 29 | 268.1 |
| OCHuman [67] | 2.5K | ITW | Real | EFT | - | 307.1 | 28 | 263.3 | 24 | 279.3 | 30 | 293.4 | 30 | 281.7 | 30 | 285.0 |
| LSPET [23] | 2.9K | ITW | Real | EFT | - | 365.7 | 31 | 292.6 | 29 | 340.1 | 31 | 339.8 | 31 | 316.3 | 31 | 330.9 |
| SSP3D [53] | 311 | ITW | Real | Yes | - | 549.8 | 32 | 522.4 | 32 | 548.1 | 32 | 439.0 | 32 | 539.5 | 32 | 519.8 |

**Evaluation Metrics.** We use standard metrics for EHPS. PVE (per-vertex error) and MPJPE (mean per-joint position error) measure the mean L2 error for vertices and regressed joints, respectively. The "PA" prefix indicates Procrustes Alignment is conducted before error computation. AGORA Leaderboard [48] introduces NMVE (normalized mean vertex error) and NMJE (normalized mean joint error) that take detection performance F1 score into consideration. Moreover, we propose MPE (mean primary error) that takes the mean of multiple primary metrics (MPJPE for 3DPW [56] test, and PVE for AGORA, UBody, EgoBody, and EHF) to gauge generalizability. All errors are reported in millimeters (mm).

## 3.2 Overview of Data Sources

In this work, we study three major types of datasets. 1) motion capture datasets that leverage optical [21, 14, 12, 15, 16, 42] or vision-based [66, 17, 7, 5] multi-view motion capture systems, are typically collected in a studio environment. However, it is possible to include an outdoor setup, or utilize additional sensors such as IMUs [56]. These datasets generally provide high-quality 3D annotations but are less flexible due to physical constraints, especially those built with immobile capture systems that require accurate sensor calibrations. 2) pseudo-annotated datasets [38, 1, 34, 37, 43, 27, 67, 2, 23, 46, 62, 53] that re-annotate existing image datasets with parametric human annotations [24, 45, 37]. These datasets take advantage of the diversity of 2D datasets, and the pseudo-3D annotations, albeit typically not as high-quality, have been proven effective [47, 31, 24]. 3) synthetic datasets [4, 6, 30, 48, 61] that are produced with renderings engines (*e.g.*, Unreal Engine). These datasets produce the most accurate 3D annotations and can easily scale up with high diversity. However, the synthetic-real gap is not fully addressed. Key attributes of the datasets are included in Table 1.

To evaluate the EHPS capability across diverse scenarios, we select multiple key datasets to form a comprehensive benchmark. They should possess the desirable traits such as 1) having accurate SMPL or SMPL-X annotations, 2) being representative of certain aspects of real-life scenarios, 3) being widely used, but this requirement is relaxed for the new datasets which are released within two years, and 4) has a clearly defined test set. To this end, five datasets (AGORA [48], UBody [37], EgoBody [66], 3DPW [56], and EHF [49]) representing different aspects are selected as the evaluation datasets. We briefly introduce these five datasets and the rest in the Supplementary Material. **AGORA**

is the most widely-used benchmark for SMPL-X evaluation. It is a synthetic dataset featuring diverse subject appearances, poses, and environments with high-quality annotation. We evaluate on both validation and test set (leaderboard) as the latter has a monthly limit of submissions. **UBody** is the latest large-scale dataset with pseudo-SMPL-X annotations that covers fifteen real-life scenarios, such as talk shows, video conferences, and vlogs, which primarily consist of the upper body in images. We follow the intra-scene protocol in training and testing, where all scenarios are seen. **EgoBody** captures human motions in social interactions in 3D scenes with pseudo-SMPL-X annotations. It comprises a first-person egocentric set (EgoSet) and a third-person multi-camera set (MVSet). We test on the EgoSet with heavy truncation and invisibility. **3DPW** is the most popular in-the-wild dataset with SMPL annotations. Since SMPL-X annotation is not available, we map SMPL-X keypoints and test on 14 LSP [22] keypoints following the conventional protocol [26, 31]. **EHF** is a classic dataset with 100 curated frames of one subject in an indoor studio setup, with diverse body poses and especially hand poses annotated in SMPL-X vertices. It has a test set but no training or validation sets. Hence, it is only used to evaluate cross-dataset performance.

Besides being popular or the latest evaluation sets for EHPS, we further analyze if these five datasets collectively provide wide coverage of existing datasets. In Fig. 3, we randomly downsample all datasets to equal length (1K examples) and employ UMAP [41] to visualize several key aspects. We use pretrained ViT-L from HumanBench [55] and OSX [37] to process patch tokens flattened as feature vectors from images cropped by bounding boxes. HumanBench is trained for various human-centric tasks (*e.g.*, Re-ID, part segmentation, and 2D pose estimation), whereas OSX is an expert model on EHPS. As for global orientation, it is closely associated with camera pose as we convert all data into the camera coordinate frame; we plot its distribution by using flattened rotation matrix representations. Moreover, we follow [50, 6, 47] to represent poses as 3D keypoints regressed from the parametric model. Specifically, we flatten 21 SMPL-X body keypoints, and 15 hand keypoints from each hand, regressed with zero parameters except for the body pose and hand poses. It is shown that 1) the five benchmark datasets have varied distribution, which is expected due to their different designated purposes, and 2) collectively, the five datasets provide a wide, near-complete coverage of the entire dataset pool.

### 3.3 Benchmarking on Individual Datasets

In this section, we aim to benchmark datasets and find those that do well in various scenarios. To gauge the performance of each dataset, we train a SMPLer-X model with the training set of that dataset and evaluate the model on the *val/testing* sets of five evaluation datasets: AGORA, UBody, EgoBody, 3DPW, and EHF. Here, the benchmarking model is standardized to use ViT-S as the backbone, trained on 4 V100 GPUs for 5 epochs with a total batch size of 128 and a learning rate of $1 \times 10^{-5}$. The dataset preprocessing details are included in the Supplementary Material.

In Table 1, we report the primary metrics (Sec. 3.1) and ranking of the 32 datasets. The complete results in the Supplementary Material. We also compute the mean primary error (MPE) to facilitate easy comparison between individual datasets. Note that for AGORA, UBody, EgoBody, and 3DPW, their performances on their own test set are excluded from computing MPE. This is because in-domain evaluation results are typically much better than cross-domain ones, leading to significant error drops. In addition, note that there are datasets designed for specific purposes (*e.g.*, Talkshow [62] for gesture generation, DNA-Rendering [7] for human NeRF reconstruction), being ranked lower on our benchmark, which focuses on EHPS (a perception task) does not reduce their unique values and contributions to the computer vision community.

From the benchmark, we observe models trained on a single dataset tend to perform well on the same domain but often cannot do well on other domains. For example, the model trained on AGORA is ranked $1^{st}$ on AGORA (val), but $6^{th}$ on UBody, $6^{th}$ on EgoBody, $12^{th}$ on 3DPW, and $24^{th}$ on EHF. This observation indicates that 1) the test scenarios are diverse, showcasing the challenging landscape of EHPS, and 2) data scaling is essential for training a robust and transferable model for EHPS due to significant gaps between different domains.

### 3.4 Analyses on Dataset Attributes

In this section, we study attributes that contribute to generalizability. However, it is important to acknowledge that such analyses are not a straightforward task: the attributes often exhibit coupled

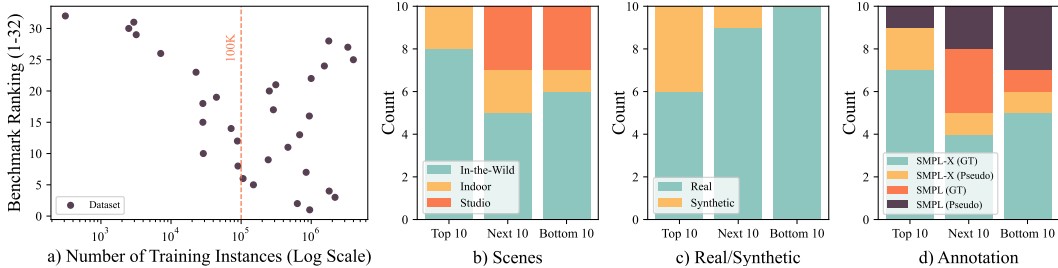

Figure 3: **Analysis on dataset attributes.** We study the impact of a) the number of training instances, b) scenes, c) real or synthetic appearance, and d) annotation type, on dataset ranking in Table 1.

effects. Consequently, counter-examples are inevitable (*e.g.*, we observe that InstaVariety, an in-the-wild dataset, demonstrates strong performance, whereas LSPET, another in-the-wild dataset, does not perform as well). Despite the challenges in pinpointing the exact factors that determine the success of an individual dataset, we adopt a collective perspective and aim to identify general trends with several key factors [47, 39, 6] in Fig. 3, and discussed below.

**First**, Fig. 3a) shows that the performance of a dataset (in terms of ranking) is not strongly associated with the number of training instances once the instance number exceeds approximately 100K. Although a very small amount of training data is insufficient to train a strong model, having an exceedingly large amount of data does not guarantee good performance either. For example, MSCOCO only comprises 149.8K training instances but achieves a higher ranking compared to datasets with $10\times$ larger scales. This may be attributed to the diverse appearance and complex scenes present in the MSCOCO dataset. Hence, it would be more cost-effective to channel resources to improve diversity and quality, when the dataset has become adequately large.

**Second**, we categorize datasets into 1) in-the-wild, which contains data from diverse environments; 2) indoor with several scenes; 3) studio, which has a fixed multi-view setup. Particularly, Fig. 3b) shows that the top 10 are mostly in-the-wild datasets, indoor datasets concentrate in the top 20 and the studio dataset tends to be ranked lower in the benchmark. Moreover, Fig. 2e) illustrates that in-the-wild datasets exhibit the most diverse distribution, covering both indoor and studio datasets. Indoor datasets display a reasonable spread, and studio datasets have the least diversity. Our findings validate previous studies that suggest an indoor-outdoor domain gap [25]. Differing from Pang *et al.* [47], which does not differentiate between indoor and studio datasets, we argue that categorizing all datasets collected indoors into a single class oversimplifies the analysis. For example, consider EgoBody [66] and Human3.6M [21]. Both datasets does not have outdoor data; however, EgoBody consists of a wide variety of indoor scenes, whereas Human3.6M consists of only one scene, which may contribute to the better ranking of EgoBody compared to Human3.6M. Hence, this suggests that in-the-wild data collection is the most ideal, but diversifying indoor scenes is the best alternative.

**Third**, most of the five contemporary synthetic datasets [4, 61, 48, 30, 6] demonstrate surprising strength and are ranked highly in Fig. 3c). It is worth noting that four (UBody, EgoBody, 3DPW, and EHF) of the five evaluation benchmarks used are real datasets, indicating that knowledge learned from synthetic data is transferable to real scenarios. To explain this observation, we take a close look at Fig. 2f): although real and synthetic datasets do not have extensive overlap, synthetic data possesses two ideal characteristics. First, there is a high overlap between real and synthetic data at the rightmost cluster. Referring to Fig. 2e), which is drawn from the same distribution, we find that this cluster primarily represents in-the-wild data. Therefore, synthetic data includes a substantial number of in-the-wild images that closely resemble real in-the-wild scenarios. Second, synthetic data also have scatters of image features on other clusters, indicating that synthetic data provides coverage to some extent for various real-world scenarios.

**Fourth**, Fig. 3d) reveals that a dataset can be valuable with accurate or pseudo-SMPL-X annotations, as they constitute the most of the top 10 datasets. A prominent example is InstaVariety [27], which has only pseudo-SMPL-X annotation produced by NeuralAnnot [45], yet, is ranked third in our benchmark. However, due to the differences in parameter spaces, SMPL annotations are less effective: it is observed that datasets with SMPL annotations tend to cluster in the lower bracket of the benchmark, especially those with pseudo-SMPL annotations. This observation suggests that SMPL-X

Table 2: **Foundation Models.** We study the scaling law of the amount of data and the model sizes. The metrics are MPJPE for 3DPW, and PVE for other evaluation benchmarks. Foundation models are named "SMPLer-X-MN", where M indicates the size of ViT backbone (S, B, L, H), N is the number of datasets used in the training. FPS: inference speed (frames per second) on a V100 GPU. MPE: mean primary error. AGORA uses the validation set, and EgoBody uses the EgoSet.

| #Datasets | #Inst. | Model | #Param. | FPS | AGORA [48] | EgoBody [66] | UBody [37] | 3DPW [56] | EHF [49] | MPE |
|---|---|---|---|---|---|---|---|---|---|---|
| 5 | 0.75M | SMPLer-X-S5 | 32M | 36.2 | 119.0 | 114.2 | 110.1 | 110.2 | 100.5 | 110.8 |
| 10 | 1.5M | SMPLer-X-S10 | 32M | 36.2 | 116.0 | 88.6 | 107.7 | 97.4 | 89.9 | 99.9 |
| 20 | 3.0M | SMPLer-X-S20 | 32M | 36.2 | 109.2 | 84.3 | 70.7 | 87.5 | 86.6 | 87.7 |
| 32 | 4.5M | SMPLer-X-S32 | 32M | 36.2 | 105.2 | 82.5 | 68.1 | 83.2 | 74.1 | 82.6 |
| 5 | 0.75M | SMPLer-X-B5 | 103M | 33.1 | 102.7 | 108.1 | 105.8 | 104.8 | 96.1 | 103.5 |
| 10 | 1.5M | SMPLer-X-B10 | 103M | 33.1 | 97.8 | 76.4 | 107.3 | 89.9 | 74.7 | 89.2 |
| 20 | 3.0M | SMPLer-X-B20 | 103M | 33.1 | 95.6 | 75.5 | 65.3 | 83.5 | 73.0 | 78.6 |
| 32 | 4.5M | SMPLer-X-B32 | 103M | 33.1 | 88.0 | 72.7 | 63.3 | 80.3 | 67.3 | 74.3 |
| 5 | 0.75M | SMPLer-X-L5 | 327M | 24.4 | 88.3 | 98.7 | 110.8 | 97.8 | 89.5 | 97.0 |
| 10 | 1.5M | SMPLer-X-L10 | 327M | 24.4 | 82.6 | 69.7 | 104.0 | 82.5 | 64.0 | 80.6 |
| 20 | 3.0M | SMPLer-X-L20 | 327M | 24.4 | 80.7 | 66.6 | 61.5 | 78.3 | 65.4 | 70.5 |
| 32 | 4.5M | SMPLer-X-L32 | 327M | 24.4 | 74.2 | 62.2 | 57.3 | 75.2 | 62.4 | 66.2 |
| 5 | 0.75M | SMPLer-X-H5 | 662M | 17.5 | 89.0 | 87.4 | 102.1 | 88.3 | 68.3 | 87.0 |
| 10 | 1.5M | SMPLer-X-H10 | 662M | 17.5 | 81.4 | 65.7 | 100.7 | 78.7 | **56.6** | 76.6 |
| 20 | 3.0M | SMPLer-X-H20 | 662M | 17.5 | 77.5 | 63.5 | 59.9 | **74.4** | 59.4 | 67.0 |
| 32 | 4.5M | SMPLer-X-H32 | 662M | 17.5 | **69.5** | **59.5** | **54.5** | 75.0 | 56.8 | **63.1** |

annotations are critical to EHPS; fitting pseudo labels is a useful strategy even if they could be noisy. Moreover, using SMPL labels effectively for SMPL-X estimation remains a challenge.

# 4 Scaling up EHPS

## 4.1 Model Architectures

Catering to our investigation, we design a minimalistic framework (dubbed SMPLer-X) that only retains the most essential parts for two reasons. First, it must be scalable and efficient as we train with a large amount of data. Second, we aim to create a framework that is decoupled from specific algorithm designs, providing a clean foundation for future research. To this end, SMPLer-X consists of three parts: a *backbone* extracts image features, which we employ Vision Transformer [11] for its scalability; a *neck* that predicts bounding boxes and crop regions of interest from the feature map for hands and face; regression *heads* that estimate parameters for each part. Note that SMPLer-X does not require third-party detectors [51], cross-part feature interaction modules [8, 13], projection of coarse SMPL-X estimations [63], or a heavy decoder [37]. As the design of SMPLer-X is not the focus of our investigation, more details are included in the Supplementary Material.

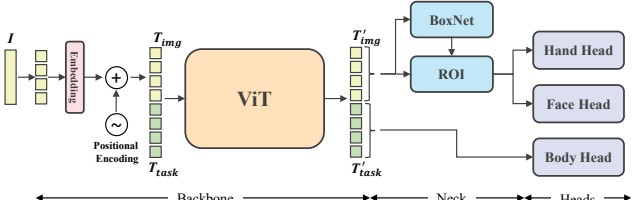

Figure 4: **Architecture of SMPLer-X**, which upholds the idea that "simplicity is beauty". SMPLer-X contains a backbone that allows for easy investigation on model scaling, a neck for hand and face feature cropping, and heads for different body parts. Note that we wish to show in this work that model and data scaling are effective, even with a straightforward architecture.

## 4.2 Training the Generalist Foundation Models

The SOTA methods [37, 44] usually train with only a few (*e.g.*, MSCOCO, MPII, and Human3.6M) datasets, whereas we investigate training with many more datasets. However, we highlight that the dataset benchmark in Table 1 cannot be used: selecting datasets based on their performance on the test sets of the evaluation benchmarks leaks information about the test sets. Hence, we construct another dataset benchmark in the Supplementary Material, that ranks individual datasets on the *training* set of the major EHPS benchmarks. We use four data amounts: 5, 10, 20, and 32 datasets as the training set, with a total length of 0.75M, 1.5M, 3.0M, and 4.5M instances. We always prioritize higher-ranked

Table 3: **AGORA test set.** † denotes the methods that are finetuned on the AGORA training set. ∗denotes the methods that are trained on AGORA training set only.

| Method | NMVE↓ (*mm*) | | NMJE↓ (*mm*) | | MVE↓ (*mm*) | | | | | MPJPE↓ (*mm*) | | | | |
|---|---|---|---|---|---|---|---|---|---|---|---|---|---|---|
| | All | Body | All | Body | All | Body | Face | LHand | RHand | All | Body | Face | LHand | RHhand |
| BEDLAM [4] | 179.5 | 132.2 | 177.5 | 131.4 | 131.0 | 96.5 | 25.8 | 38.8 | 39.0 | 129.6 | 95.9 | 27.8 | 36.6 | 36.7 |
| Hand4Whole [44]† | 144.1 | 96.0 | 141.1 | 92.7 | 135.5 | 90.2 | 41.6 | 46.3 | 48.1 | 132.6 | 87.1 | 46.1 | 44.3 | 46.2 |
| BEDLAM [4]† | 142.2 | 102.1 | 141.0 | 101.8 | 103.8 | 74.5 | 23.1 | 31.7 | 33.2 | 102.9 | 74.3 | 24.7 | 29.9 | 31.3 |
| PyMaF-X [63]† | 141.2 | 94.4 | 140.0 | 93.5 | 125.7 | 84.0 | 35.0 | 44.6 | 45.6 | 124.6 | 83.2 | 37.9 | 42.5 | 43.7 |
| OSX [37] ∗ | 130.6 | 85.3 | 127.6 | 83.3 | 122.8 | 80.2 | 36.2 | 45.4 | 46.1 | 119.9 | 78.3 | 37.9 | 43.0 | 43.9 |
| HybrIK-X [33] | 120.5 | 73.7 | 115.7 | 72.3 | 112.1 | 68.5 | 37.0 | 46.7 | 47.0 | 107.6 | 67.2 | 38.5 | 41.2 | 41.4 |
| SMPLer-X-L20 | 133.1 | 88.1 | 128.9 | 84.6 | 123.8 | 81.9 | 37.4 | 43.6 | 44.8 | 119.9 | 78.7 | 39.5 | 41.4 | 44.8 |
| SMPLer-X-L32 | 122.8 | 80.3 | 119.1 | 77.6 | 114.2 | 74.7 | 35.1 | 41.3 | 42.2 | 110.8 | 72.2 | 36.7 | 39.1 | 40.1 |
| SMPLer-X-L20† | 107.2 | 68.3 | 104.1 | 66.3 | 99.7 | 63.5 | 29.9 | 39.1 | 39.5 | 96.8 | 61.7 | 31.4 | 36.7 | 37.2 |

Table 4: **AGORA Val set**. † and ∗ are finetuned on the AGORA training set, and trained on the AGORA training set only, respectively.

| Method | PA-PVE↓ (*mm*) | | | PVE↓ (*mm*) | | |
|---|---|---|---|---|---|---|
| | All | Hands | Face | All | Hands | Face |
| Hand4Whole [44]† | 73.2 | 9.7 | 4.7 | 183.9 | 72.8 | 81.6 |
| OSX [37] | 69.4 | 11.5 | 4.8 | 168.6 | 70.6 | 77.2 |
| OSX [37]∗ | 45.0 | 8.5 | 3.9 | 79.6 | 48.2 | 37.9 |
| SMPLer-X-B1∗ | 48.9 | 8.6 | 4.0 | 86.1 | 51.5 | 41.2 |
| SMPLer-X-L20 | 48.6 | 8.9 | 4.0 | 80.7 | 51.0 | 41.3 |
| SMPLer-X-L32 | 45.1 | 8.7 | 3.8 | 74.2 | 47.8 | 38.7 |
| SMPLer-X-L20† | 39.1 | 9.3 | 3.8 | 62.5 | 42.3 | 32.8 |

Table 5: **EHF**. As EHF does not have a training set to benchmark datasets, we do not perform finetuning. Moreover, EHF is not seen in our training and can be used to validate our foundation models' transferability.

| Method | PA-PVE↓ (*mm*) | | | PVE↓ (*mm*) | | |
|---|---|---|---|---|---|---|
| | All | Hands | Face | All | Hands | Face |
| Hand4Whole [44] | 50.3 | 10.8 | 5.8 | 76.8 | 39.8 | 26.1 |
| OSX [37] | 48.7 | 15.9 | 6.0 | 70.8 | 53.7 | 26.4 |
| SMPLer-X-L20 | 37.8 | 15.0 | 5.1 | 65.4 | 49.4 | 17.4 |
| SMPLer-X-L32 | 37.1 | 14.1 | 5.0 | 62.4 | 47.1 | 17.0 |

datasets. To prevent larger datasets from shadowing smaller datasets, we adopt a balanced sampling strategy. Specifically, all selected datasets are uniformly upsampled or downsampled to the same length and add up to the designated total length. To facilitate training, we follow OSX [37] to use AGORA, UBody, MPII, 3DPW, Human3.6M in COCO-format [38], and standardize all other datasets into the HumanData [10] format. We also study four ViT backbones of different sizes (ViT-Small, Base, Large and Huge), pretrained by ViTPose [60]. The training is conducted on 16 V100 GPUs, with a total batch size of 512 (256 for ViT-Huge) for 10 epochs. More training details such as adapting SMPL or gendered SMPL-X in the training are included in the Supplementary Material.

In Table 2, we show experimental results with a various number of datasets and foundation model sizes. Foundation models are named "SMPLer-X-MN", where M can be S, B, L, H that indicates the size of the ViT backbone, and N indicates the number of datasets used in the training. For example, SMPLer-X-L10 means the foundation model takes ViT-L as the backbone, and is trained with Top 10 datasets (ranked according to the individual dataset performance on the training sets of the key evaluation benchmarks). It is observed that **1)** more training data (data scaling) leads to better performance in terms of MPE. The model performance improves gradually as the number of training datasets increases. However, besides the increment in training instances, more datasets provide a richer collection of diverse scenarios, which we argue is also a key contributor to the performance gain across evaluation benchmarks. **2)** A larger foundation model (model scaling) performs better at any given amount of data. However, the marginal benefits of scaling up decrease beyond model

Table 10: **3DPW.** ‡ denotes the methods that use a head for SMPL regression. † and ∗ are finetuned on the 3DPW training set and trained on 3DPW training set only, respectively. Unit: *mm*.

| Method | MPJPE | PA-MPJPE |
|---|---|---|
| Body-only (SMPL) Methods | | |
| OSX-SMPL [37]‡∗ | 74.7 | 45.1 |
| HybrIK [35] | 71.6 | 41.8 |
| CLIFF [36] | 68.0 | 43.0 |
| Whole-Body (SMPL-X) Methods | | |
| Hand4Whole [44] | 86.6 | 54.4 |
| ExPose [8] | 93.4 | 60.7 |
| OSX [37]† | 86.2 | 60.6 |
| SMPLer-X-B1∗ | 95.6 | 67.6 |
| SMPLer-X-L20 | 78.3 | 52.1 |
| SMPLer-X-L32 | 75.2 | 50.5 |
| SMPLer-X-L20† | 76.8 | 51.5 |

size L. Specifically, ViT-H has more than twice the parameters than ViT-L, but the performance gain is not prominent. **3)** The foundation model always performs better than in-domain training on a single training set. For example, SMPLer-X-B20, performs better on the validation set of AGORA, and test sets of UBody, EgoBody, and 3DPW, than models trained specifically on the corresponding training set in Table 1. This is useful for real-life applications: instead of training a model for each of the user cases, a generalist foundation model contains rich knowledge to be a one-size-fits-all alternative.

Table 6: **UBody.** † denotes the methods that are finetuned on the UBody training set. ∗ denotes the methods that are trained on UBody training set only.

| Method | PA-PVE↓ (mm) | | | PVE↓ (mm) | | |
|---|---|---|---|---|---|---|
| | All | Hands | Face | All | Hands | Face |
| PIXIE [13] | 61.7 | 12.2 | 4.2 | 168.4 | 55.6 | 45.2 |
| Hand4Whole [44] | 44.8 | 8.9 | 2.8 | 104.1 | 45.7 | 27.0 |
| OSX [37] | 42.4 | 10.8 | 2.4 | 92.4 | 47.7 | 24.9 |
| OSX [37]† | 42.2 | **8.6** | **2.0** | 81.9 | 41.5 | 21.2 |
| SMPLer-X-B1∗ | 38.5 | 10.8 | 3.0 | 64.8 | 45.4 | 22.3 |
| SMPLer-X-L20 | 33.2 | 10.6 | 2.8 | 61.5 | 43.3 | 23.1 |
| SMPLer-X-L32 | **30.9** | 10.2 | 2.7 | **57.3** | **39.2** | 21.6 |
| SMPLer-X-L-20† | 31.9 | 10.3 | 2.8 | 57.4 | 40.2 | 21.6 |

Table 7: **EgoBody-EgoSet.** † denotes the methods that are finetuned on the EgoBody-EgoSet training set. ∗ denotes the methods that are trained on EgoBody-EgoSet training set only.

| Method | PA-PVE↓ (mm) | | | PVE↓ (mm) | | |
|---|---|---|---|---|---|---|
| | All | Hands | Face | All | Hands | Face |
| Hand4Whole [44] | 58.8 | **9.7** | 3.7 | 121.9 | 50.0 | 42.5 |
| OSX [37] | 54.6 | 11.6 | 3.7 | 115.7 | 50.6 | 41.1 |
| OSX [37]† | 45.3 | 10.0 | 3.0 | 82.3 | 46.8 | 35.2 |
| SMPLer-X-B1∗ | 56.1 | 10.7 | 3.5 | 87.2 | 49.4 | 34.9 |
| SMPLer-X-L20 | 38.9 | 9.9 | 3.0 | 66.6 | 42.7 | 31.8 |
| SMPLer-X-L32 | **36.3** | 9.8 | **2.9** | **62.2** | **41.4** | **30.7** |
| SMPLer-X-L20† | 37.8 | 9.9 | **2.9** | 63.6 | 42.5 | 30.8 |

Table 8: **ARCTIC.** † and ∗ denote the methods that are finetuned on the ARCTIC training set and trained on the ARCTIC training set only, respectively.

| Method | PA-PVE↓ (mm) | | | PVE↓ (mm) | | |
|---|---|---|---|---|---|---|
| | All | Hands | Face | All | Hands | Face |
| Hand4Whole [44] | 63.4 | 18.1 | 4.0 | 136.8 | 54.8 | 59.2 |
| OSX [37] | 56.9 | **17.5** | 3.9 | 102.6 | 56.5 | 44.6 |
| OSX [37]† | 33.0 | 18.8 | 3.3 | 58.4 | 39.4 | 30.4 |
| SMPLer-X-B1∗ | 45.2 | 18.9 | 3.4 | 66.6 | 42.5 | 34.0 |
| SMPLer-X-L10 | 46.9 | 18.1 | **2.3** | 76.9 | 50.8 | 33.2 |
| SMPLer-X-L32 | **29.4** | 18.9 | 2.7 | **48.6** | **38.8** | **26.8** |
| SMPLer-X-L10† | 33.1 | 19.0 | 2.7 | 54.9 | 40.1 | 27.3 |

Table 9: **DNA-Rendering-HiRes**. † and ∗ are finetuned on the DNA-Rendering-HiRes training set and trained on the DNA-Rendering-HiRes training set only, respectively.

| Method | PA-PVE↓ (mm) | | | PVE↓ (mm) | | |
|---|---|---|---|---|---|---|
| | All | Hands | Face | All | Hands | Face |
| Hand4Whole [44] | 62.8 | 11.0 | 4.2 | 111.4 | 56.4 | 52.6 |
| OSX [37] | 59.9 | 10.6 | 4.3 | 105.7 | 55.0 | 52.5 |
| OSX [37]† | 43.5 | 7.5 | 3.5 | 67.1 | 43.3 | 38.2 |
| SMPLer-X-B1∗ | 45.6 | 7.5 | 3.4 | 63.2 | 40.7 | 34.2 |
| SMPLer-X-L20 | 44.4 | 11.1 | 4.5 | 77.7 | 47.5 | 43.2 |
| SMPLer-X-L32 | **35.8** | **7.2** | **3.2** | **54.4** | **36.7** | **34.0** |
| SMPLer-X-L20† | 37.9 | 7.3 | 3.4 | 56.5 | 38.4 | 34.9 |

Besides the errors on key benchmarks, we also report the inference speed (in terms of FPS, or frames per second) of the SMPLer-X model family in Table 2. The testing is conducted on a single V100 GPU with batch size 1, excluding data loading. SMPLer-X family is faster than OSX (12.2 FPS on a single V100 GPU) using the same test setting, and the smaller versions such as SMPLer-X-S and SMPLer-X-B can achieve real-time performance, with SMPLer-X-L on the verge of achieving real-time speeds. The high inference speed is attributed to the minimalistic architecture of SMPLer-X, which only retains the most essential components for EHPS.

Moreover, we show detailed by-part results of body, hands, and face on main benchmarks such as AGORA test set (Table 3), AGORA validation set (Table 4), UBody (Table 6), EgoBody-EgoSet (Table 7) and EHF (Table 5). We also compare our results with whole-body methods on 3DPW (Table 10). We highlight that the foundation models show strong and balanced performances on all benchmarks.

Furthermore, we evaluate the transferability of our foundation models on two more benchmarks: ARCTIC (Table 8) and DNA-Rendering (Table 8). ARCTIC features complicated hand-object interaction with whole-body annotations, and DNA-Rendering includes diverse subjects, motions, and garments. Note that ARCTIC is not seen by foundation models trained on Top 10 datasets, and DNA-Rendering is not seen by foundation models trained on Top 20 datasets. The foundation models, however, achieve much better performance than SOTAs with conventional data sampling strategies.

In addition, we compare our foundation model with SOTA methods, such as Hand4Whole [44] and OSX [37] in various scenarios in Fig. 5. These scenarios feature challenging aspects such as heavy truncation (from only half of the body visible to only the arms visible), difficult body poses in diverse backgrounds, and rare camera angles (extremely high or low elevation angles). SMPLer-X demonstrates the strength of massive training data and consistently produces robust estimations.

### 4.3 Finetuning the Specialists

Training the foundation model with a large number of data is expensive. For example, SMPLer-X-H20 takes more than 400 GPU hours to train. Hence, it is critical to investigate finetuning strategies that allow for low-cost adaptation of the foundation model to specific scenarios. We reiterate that in real-life applications, the *test set* is inaccessible. Hence, we use our benchmarking strategy and select five high-ranking datasets on the target *training set* to finetune the model for 5 epochs. We perform

GT   Hand4Whole   OSX   SMPLer-X-L32   GT   Hand4Whole   OSX   SMPLer-X-L32

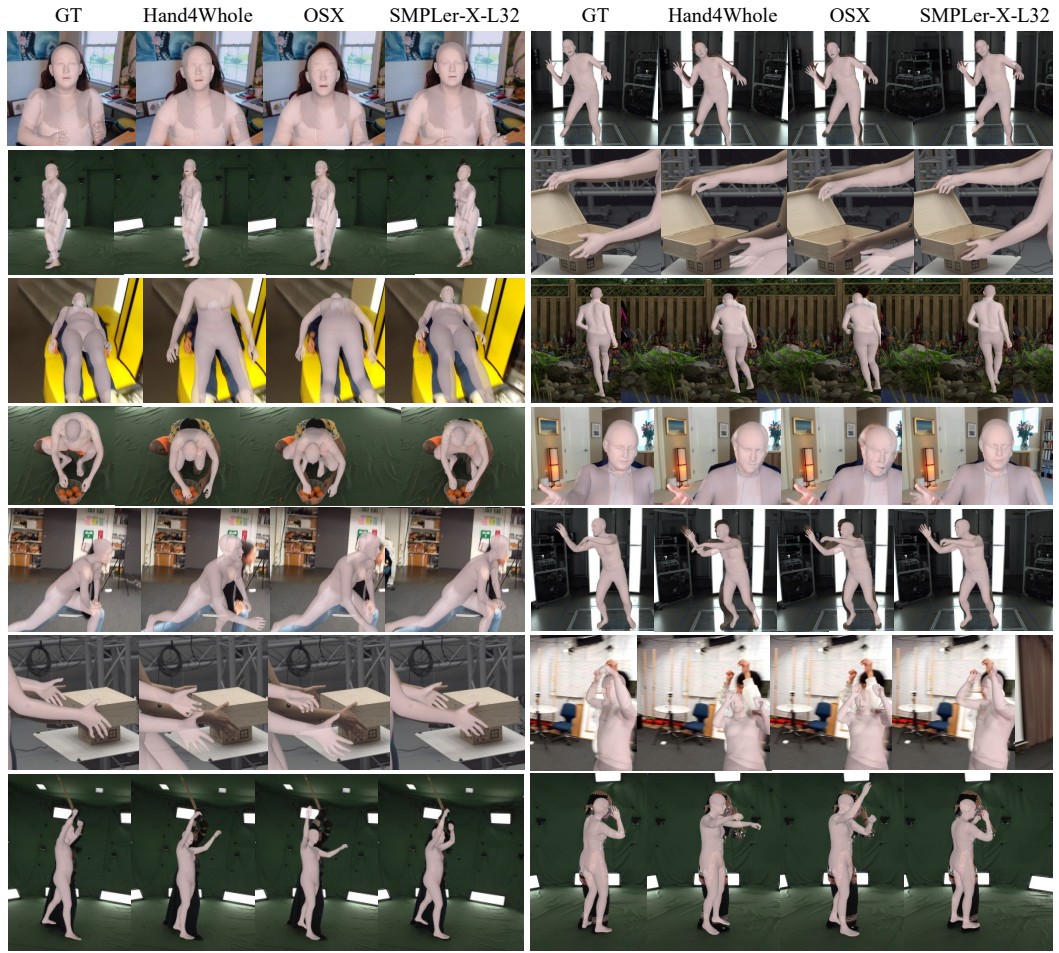

Figure 5: **Visualization.** We compare SMPLer-X-L32 with OSX [37] and Hand4Whole [44] (trained with the MSCOCO, MPII, and Human3.6M) in various scenarios such as those with heavy truncation, hard poses, and rare camera angles.

finetune experiments on ViT-L to match the backbone of current SOTA [37]. The results are shown in the same tables as the foundation models (Table 3, 4, 5, 6, 7, and 10), where finetuning always lead to substantial performance enhancement on the foundation models.

## 5   Conclusion

In this work, we benchmark datasets for EHPS that provide us insights for training and finetuning a foundation model. Our work is useful in three ways. First, our pretrained model (especially the backbone) can be a plug-and-play component of a larger system for EHPS and beyond. Second, our benchmark serves to gauge the performances of future generalization studies. Third, our benchmarking-finetuning paradigm can be useful for the rapid adaptation of any foundation model to specific scenarios. Specifically, users may collect a training set, evaluate pretrained models of various other datasets on it, and select the most relevant datasets to finetune a foundation model.

**Limitations.** First, although we use five comprehensive benchmark datasets to gauge the generalization capability, they may still be insufficient to represent the real-world distribution. Second, our experiments do not fully investigate the impact of various model architectures due to the prohibitive cost of training the foundation model.

**Potential negative societal impact.** As we study training strong EHPS models and release the pretrained models, they may be used for unwarranted surveillance or privacy violation.

## Acknowledgement

This study is supported under the RIE2020 Industry Alignment Fund – Industry Collaboration Projects (IAF-ICP) Funding Initiative, as well as cash and in-kind contribution from the industry partner(s). The project is also supported by NTU NAP and Singapore MOE AcRF Tier 2 (MOET2EP20221-0012).

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
