# SMPLer-X: Scaling Up Expressive Human Pose and Shape Estimation – Supplementary Material –

**Zhongang Cai**[*,1,2,3], **Wanqi Yin**[*,2,4], **Ailing Zeng**[5], **Chen Wei**[2], **Qingping Sun**[2],
**Yanjun Wang**[2], **Hui En Pang**[1,2], **Haiyi Mei**[2], **Mingyuan Zhang**[1],
**Lei Zhang**[5], **Chen Change Loy**[1], **Lei Yang**[†,2,3], **Ziwei Liu**[†,1]

[1] S-Lab, Nanyang Technological University, [2] SenseTime Research, [3] Shanghai AI Laboratory,
[4] The University of Tokyo, [5] International Digital Economy Academy (IDEA)

## A   Overview

Due to space constraints in the main paper, we elaborate the following here: additional details of the 32 datasets, including useful links to find their license statements and other ethics concerns in Sec. B; additional details of the architecture, training and finetuning stages in Sec. C; additional experiments and analyses on dataset distributions, training schemes, finetuning strategies, sampling strategies, and training domains in Sec. D; individual dataset ranking on the training sets of key evaluation benchmarks, and complete results of the foundation models on evaluation benchmarks in Sec. E.

## B   Additional Details of Datasets

### B.1   Dataset Descriptions

This section describes the 32 datasets we study. Note that all these are public academic datasets, each holding a license. We follow the common practice to use them in our non-commercial research and refer readers to their homepages or papers for more details regarding *licenses* and their policies to ensure personal information protection.

**3DPW** [37] (Fig. 1a) is the first in-the-wild dataset with a considerable amount of data, captured with a moving phone camera and IMU sensors. It features accurate SMPL annotations and 60 video sequences captured in diverse environments. We follow the official definition of train, val, and test splits. Homepage: `https://virtualhumans.mpi-inf.mpg.de/3DPW/`.

**AGORA** [34] (Fig. 1b) is a synthetic dataset, rendered with high-quality human scans and realistic 3D scenes. It consists of 4240 textured human scans with diverse poses and appearances, each fitted with accurate SMPL-X annotations. There are 14K training images and 3K test images, and 173K instances. Homepage: `https://agora.is.tue.mpg.de/index.html`

**ARCTIC** [12] (Fig. 1c) is a lab-based hand-object interaction dataset. It features 10 subjects manipulating 11 objects. There are 210K frames of video sequences captured from 8 static cameras and one egocentric camera. Each frame is fitted with accurate SMPL-X annotations. We exclude the egocentric frames in our training as they only capture hands, and use 153.9K images in training. Homepage: `https://arctic.is.tue.mpg.de/`

**BEDLAM** [5] (Fig. 1d) is a synthetic dataset that includes a wide range of variations in terms of body shapes, motions, skin tones, hair, and clothing. It is created by combining 271 different body models, 27 hairstyles, and 111 types of clothing. The dataset includes 1691 clothing textures and

---

[*]Equal contributions. [†]Co-corresponding authors.

37th Conference on Neural Information Processing Systems (NeurIPS 2023) Track on Datasets and Benchmarks.

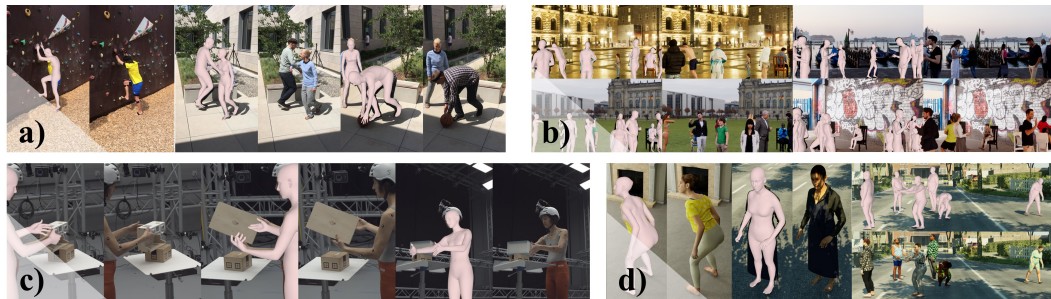

Figure 1: **Visualization** of dataset images and ground truth annotation. a) 3DPW. b) AGORA. c) ARCTIC. d) BEDLAM.

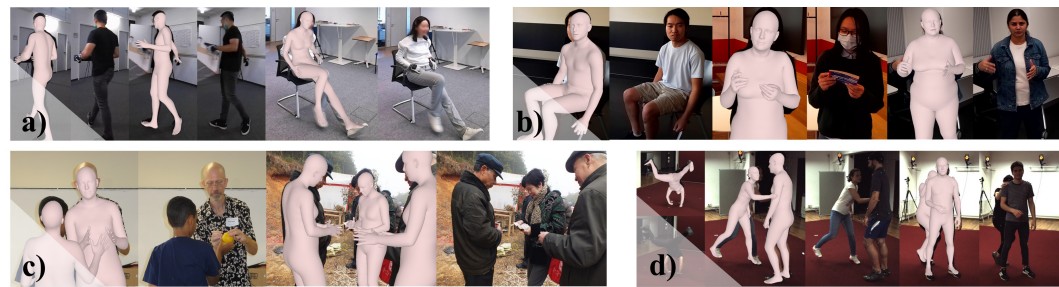

Figure 2: **Visualization** of dataset images and ground truth annotation. a) BEHAVE. b) EgoBody (EgoSet). c) CrowdPose. d) CHI3D.

2311 human motions set in 95 HDRI and 8 3D scenes. Each scene typically consists of 1 to 10 people and offers diverse camera poses. Homepage: `https://bedlam.is.tue.mpg.de/index.html`

**BEHAVE** [3] (Fig. 2a) is a body human-object interaction dataset with multi-view RGB-D frames, SMPL-H parameters, object fits, and contacts information. BEHAVE includes about 15k frames in 5 locations with 8 subjects performing a range of interactions with 20 common objects. Homepage: `https://github.com/xiexh20/behave-dataset`.

**CHI3D** [13] (Fig. 2d) is a studio-based 3D motion capture dataset (Vicon) under multiple interaction scenarios, which includes 631 multi-view sequences with 2,525 contact events and 728,664 ground truth instances of 3D poses annotated with SMPL-X parameters. We use the open-source train set. Homepage: `https://ci3d.imar.ro`.

**CrowdPose** [25] (Fig. 2c) is an in-the-wild dataset focused on crowded cases. It contains 20K images in total and 80K human instances. In this paper, we use the annotations generated by NeuralAnnot [32], which fits the SMPL to the GT 2D joints and includes a total of ~35.7K annotated data. Homepage: `https://github.com/Jeff-sjtu/CrowdPose`

**EgoBody** [40] is a large-scale dataset that features 3D human motions and interaction with scenes. The data is captured by a multi-view rig for third-person view (MVSet, in Fig. 3a) and a head-mounted device for egocentric view (EgoSet, in Fig. 2b). The dataset consists of 125 sequences, 36 subjects, and 15 indoor scenes. We follow the official splits of training and test sets. Homepage: `https://sanweiliti.github.io/egobody/egobody.html`.

**EHF** [35] (Fig. 3b) contains 100 curated frames of one subject in an indoor studio setup. It provides SMPL-X aligned 3D mesh as the ground truth that accurately reflects the subject' diverse body, hand, and face articulations. It is usually used as a test set. The images are captured from a single camera. It is published along with SMPL-X. Homepage: `https://smpl-x.is.tue.mpg.de/index.html`.

**FIT3D** [15] (Fig. 3c) is a studio-based 3D motion capture dataset including 611 multi-view sequences with 2,964,236 images and corresponding ground truth instances of 3D shapes and poses annotated with SMPL-X parameters. Motion clips include 37 repeated exercises. We use the open-source train set. Homepage: `https://fit3d.imar.ro/`.

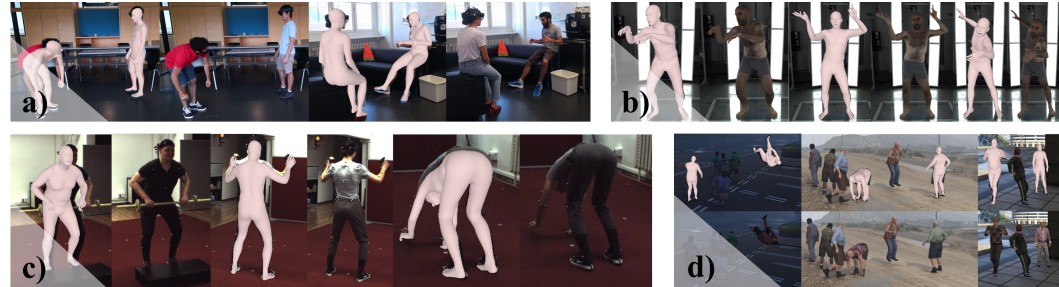

Figure 3: **Visualization** of dataset images and ground truth annotation. a) EgoBody (MVSet). b) EHF. c) FIT3D. d) GTA-Human.

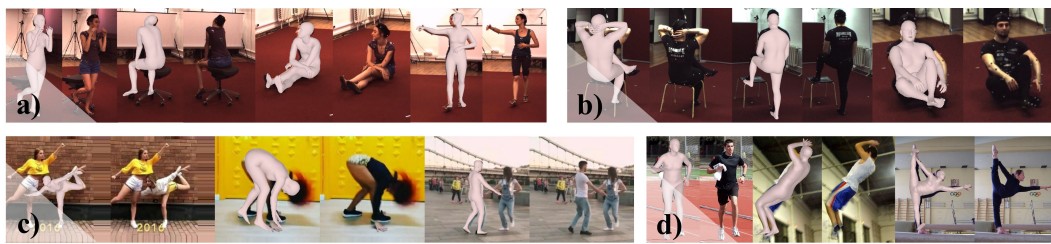

Figure 4: **Visualization** of dataset images and ground truth annotation. a) Human3.6M. b) HumanSC3D. c) InstaVariety. d) LSPET.

**GTA-Human II** (Fig. 3d) is an extended version of GTA-Human [8], a large-scale synthetic 3D single-human dataset generated with the GTA-V game engine, which features diversity. GTA-Human provides more than 1.4M of SMPL annotations in single-person scenes. In comparison, GTA-Human II includes multi-human scenarios with SMPL-X ground truth, obtained through SMPLify-X [35], which estimates SMPL-X parameters from ground truth keypoints collected in-game. The toolchain is provided by MMHuman3D [10]. The extended version contains 1.8M SMPL-X instances. Images are captured in 4K multi-person sequences, with about 600 subjects in different shapes and clothing, performing 20K daily human activity motion clips in six distinct categories of backgrounds, captured by camera angles in realistic distributions. Homepage: `https://caizhongang.github.io/projects/GTA-Human/`.

**Human3.6M** [18] (Fig. 4a) is a studio-based 3D motion capture dataset including 3.6M human poses and corresponding images captured by a high-speed motion capture system. In this paper, we use the annotation generated by NeuralAnnot [32], which fits the SMPL-X to the GT 2D joints and includes a total of ~312.2K annotated data. Homepage: `http://vision.imar.ro/human3.6m/description.php`

**HumanSC3D** [14] (Fig. 4b) is a studio-based 3D motion capture dataset including 1,032 multiple-view sequences featuring 5K contact events and 1.2M ground truth instances of 3D poses annotated with SMPL-X parameters. We use the open-source train set. Homepage: `https://sc3d.imar.ro/`.

**InstaVariety** [22] (Fig. 4c) is an in-the-wild dataset, containing 2.1M images collected from Instagram using 84 hashtags. We use the annotation generated by NeuralAnnot [32], which fits the SMPL to the GT 2D joints and includes a total of ~218.5K annotated data. Homepage: `https://github.com/akanazawa/human_dynamics/blob/master/doc/insta_variety.md`

**LSPET** [19] (Fig. 4d) is an in-the-wild dataset, and it contains 10K images. In this paper, we use the annotation generated by EFT [21], which fits the SMPL to the GT 2D joints and includes a total of 2,946 annotated data. Homepage: `http://sam.johnson.io/research/lspet.html`.

**MPI-INF-3DHP** [29] ((Fig. 5a) is captured with a multi-camera markerless motion capture system in constrained indoor and complex outdoor scenes. It records 8 actors performing 8 activities from 14 camera views. We use the annotations generated by NeuralAnnot [32], which fits the

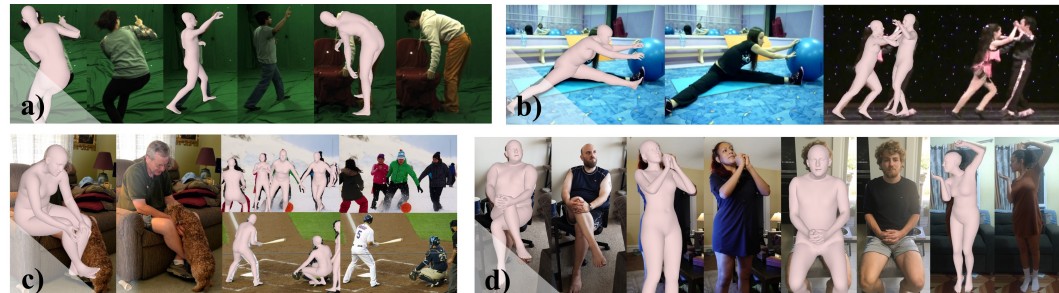

Figure 5: **Visualization** of dataset images and ground truth annotation. a) MPI-INF-3DHP. b) MPII. c) MSCOCO. d) MTP.

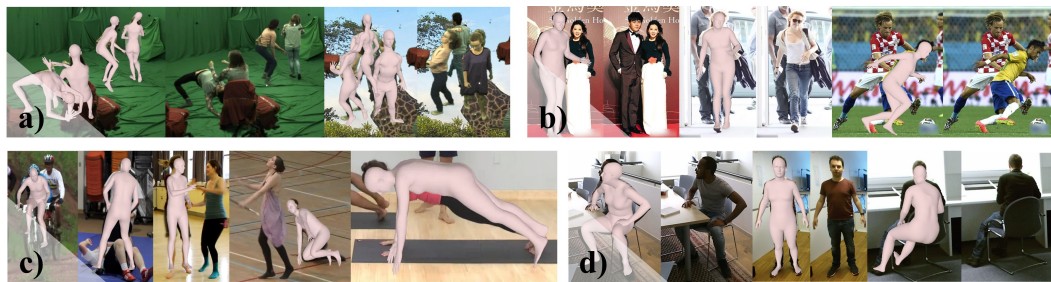

Figure 6: **Visualization** of dataset images and ground truth annotation. a) MuCo-3DHP. b) OCHuman. c) PoseTrack. d) PROX.

SMPL-X to the GT 2D joints and includes a total of 939,847 annotated data. Homepage: `https://vcai.mpi-inf.mpg.de/3dhp-dataset/`

**MPII** [2] ((Fig. 5b) is a widely used in-the-wild dataset that offers a diverse collection of approximately 25K images. Each image within the dataset contains one or more instances, resulting in a total of over 40K annotated people instances. Among the 40K samples, ~28K samples are used for training, while the remaining samples are reserved for testing. We use the annotations generated by NeuralAnnot [32], which fits the SMPL-X to the GT 2D joints and includes a total of ~28.9K annotated data. Homepage: `http://human-pose.mpi-inf.mpg.de/`

**MSCOCO** [27] (Fig. 5c) is a large-scale object detection, segmentation, keypoint detection, and captioning dataset. The subset for the keypoint detection contains more than 200K images and 250K person instances. We use the annotations generated by NeuralAnnot [32], which fits the SMPL-X to the GT 2D joints and includes a total of ~149.8K annotated data. Homepage: `https://cocodataset.org/#home`

**MTP** [33] (Fig. 5d) is an in-door dataset containing images of actors mimicking different hard SMPL-X poses with self-contact. There are 3.7K images from 148 subjects with pseudo ground-truth SMPL-X parameters and 2D keypoints. We use 3.2K instances in training. Homepage: `https://tuch.is.tue.mpg.de/`

**MuCo-3DHP** [30] (Fig. 6a) is an in-door multi-person dataset composited by cropping and overlaying person in MPI-INF-3DHP[29] with segmentation masks. It has 400K frames and contains 8 subjects with 2 different clothing for each subject. It is shot with 12 different camera positions. It has ground truth 3D keypoints and fitted SMPL parameters. We use 465.3K annotated data in training. Homepage: `https://vcai.mpi-inf.mpg.de/projects/SingleShotMultiPerson/`.

**OCHuman** [41] (Fig. 6b) is an in-the-wild datset, and it focuses on heavily occluded human. This dataset contains 8,110 detailed annotated human instances within 4,731 images. We use the annotations generated by EFT [21], which fits the SMPL to the GT 2D joints and includes a total of 2,495 annotated data. Homepage: `https://github.com/liruilong940607/OCHumanApi`

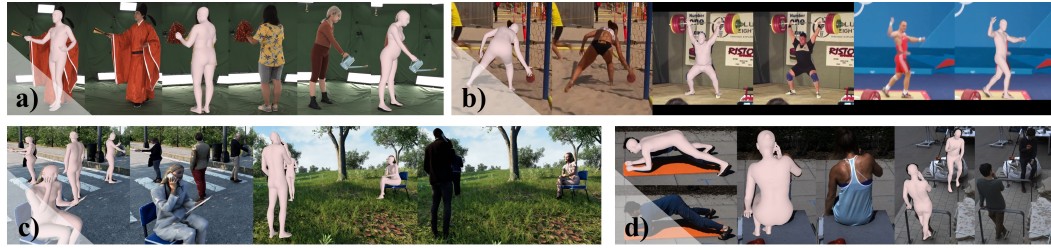

Figure 7: **Visualization** of dataset images and ground truth annotation. a) DNA-Rendering. b) SSP3D. c) SPEC. d) RICH.

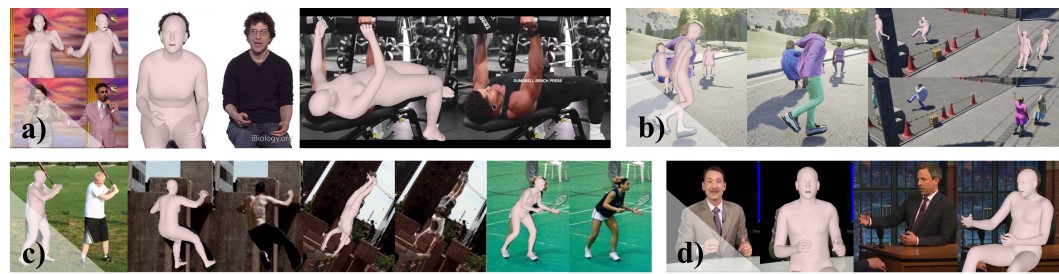

Figure 8: **Visualization** of dataset images and ground truth annotation. a) UBody. b) SynBody. c) UP3D. d) Talkshow.

**PoseTrack** [1] (Fig. 6c) is a large-scale benchmark for multi-person pose estimation and tracking in videos. It contains 514 videos and includes 66,374 frames. We use the annotations generated by EFT [21], which fits the SMPL to the GT 2D joints and includes a total of ~28.5K annotated data. Homepage: `https://posetrack.net`

**PROX** [16] (Fig. 6d) qualitative dataset is a human-scene interaction dataset that showcases 12 indoor scenes and 20 subjects engaging with these scenes. It comprises 100K RGB-D frames with pseudo-ground-truth SMPL-X fittings. During training, only the RGB images are utilized, and they are horizontally flipped to align with the SMPL-X annotations. We use 88.1K instances for training. Homepage: `https://prox.is.tue.mpg.de/`.

**DNA-Rendering** [9] (Fig. 7a) is a large-scale multi-view studio-based dataset with different resolutions (main set and HiRes set) that features diversity in motion, clothing, and object interactions. DNA-Rendering has more than 1.5K human instances and 5K motion sequences with up to 60 RGB views and 4 Kinect views. Corresponding SMPL-X annotation is based on HuMMan [7]. We separate the 60 RGB views into 48 and 12 views based on different camera distributions and captured resolutions. Homepage: `https://dna-rendering.github.io/`.

**SPEC** [23] (Fig. 7c) is a synthetic dataset featuring diverse and unique camera viewpoints. It has 22,191 images with 71,982 ground truth instances with SMPL parameters as a train set and 3,783 images with 12,071 ground truth instances as the test set. Homepage: `https://spec.is.tue.mpg. de/index.html`.

**RICH** [17] (Fig. 7d) is a human-scene contact dataset. It includes a comprehensive collection of 142 single or multi-person multiview videos capturing 22 subjects in 5 static indoor or outdoor scenes with 6-8 static cameras. RICH comprises a rich set of resources, including a total of 90K posed 3D body meshes, each associated with dense full-body contact labels in both SMPL-X and SMPL mesh topology. We convert the original image from .png and .bmp to .jpg and train the model with the train set, which includes ~243.4K instances. Homepage: `https://rich.is.tue.mpg.de/index.html`

**SSP3D** [36] SSP-3D (Fig. 7b) is a small-scale dataset consisting of 311 images of persons in tight-fitted clothes in sports, with a variety of body shapes and poses. Pseudo-ground-truth SMPL body model parameters obtained via multi-frame optimization with shape consistency. Homepage: `https://github.com/akashsengupta1997/SSP-3D`.

**SynBody** [38] (Fig. 8b) is a large-scale synthetic dataset featuring a massive number of diverse subjects and high-accuracy annotations which includes multi-person image instances with 3D pose and shape annotations. SynBody covers 10K human body models, 1K actions, and many viewpoints. Annotations include both accurate SMPL and SMPL-X parameters. Synbody also features layered human body models and clothes. We sample a set with ~600K instances in our study. Homepage: `https://maoxie.github.io/SynBody/`.

**Talkshow** [39] (Fig. 8d) is a large-scale dataset featuring talking videos of 4 subjects in 4 different scenarios. It contains 26.9 hours of video clips at 30 FPS and has synchronized audio and fitted SMPL-X annotations. We obtain the video clips from the author and convert them to images, including of 332.7K instances. Homepage: `https://talkshow.is.tue.mpg.de/`.

**UBody** [26] (Fig. 8b) is a large-scale dataset that features a diverse range of real-life scenarios that cater to various downstream tasks, such as fitness videos, VLOGs, movies, online classes, video conferences, talk shows, and sign languages. In these scenarios, typically only the subject's upper body is visible. Heavy truncation and a focus on expressive gestures and facial expressions make UBody especially challenging. Homepage: `https://github.com/IDEA-Research/OSX`.

**UP3D** [24] (Fig. 8c) is an in-the-wild dataset containing 7,126 images. To obtain 3D high-quality annotations, it extends the SMPLify [6] and fits a pseudo label (SMPL) for each image. Homepage: `https://files.is.tuebingen.mpg.de/classner/up/`

## C    Additional Details of Foundation Model

### C.1    Architecture

SMPLer-X utilizes a minimalistic design. Before entering the backbone, the image is cropped by a whole body bounding box and resized to $I$ with (height, width) as (512, 384). The image is then tokenized into $32 \times 24$ patches with patch size 16, and undergoes patch embedding, and positional encoding is added to obtain image tokens $T_{img}$. $T_{task}$ is additional learnable tokens (task tokens) that are concatenated with $T_{img}$. The tokens are processed with *backbone* (denoted as ViT). Leveraging the scalability of ViT [11], we are able to experiment with various model sizes. In the *neck*, the processed image tokens, $T'_{img}$ are used to predict face and hand bounding boxes. The predicted bounding boxes are used in the ROI (regions of interest) module to crop features from $T'_{img}$, which is re-organized and undergoes transposed convolution (deconv), and fed into hand and face heads. The body head takes in both $T'_{img}$ (omitted in the illustration) and $T'_{task}$. The hand and body heads consist of a positional module to predict 3D keypoints, and a regressor module to predict parameters, whereas for the face head, we follow OSX [26] to include only a regressor module. We highlight that training foundation models are very expensive. Hence, we do not conduct extensive architectural searches in our study. We use SMPLer-X as a simple baseline with the essential components, which (*e.g.*, backbone) can be directly used in future research. In addition, the data selection strategies in our study are likely to be applicable to any other architectures.

### C.2    Training Details

The training is conducted on 16 V100 GPUs, with a total batch size of 512 (256 for ViT-Huge) for 10 epochs. Specifically, SMPLer-X-L20 takes more than 400 GPU hours to train and SMPLer-X-H32 takes more than 700 GPU hours to train. We use Adam optimizer with cosine annealing for both training and fine-tuning. The learning rate for training is $1 \times 10^{-5}$ with the minimum learning rate set to $1 \times 10^{-6}$, while the learning rate for finetuning is $1 \times 10^{-5}$ with the minimum learning rate set to $5 \times 10^{-7}$.

### C.3    Adaption of SMPL/SMPL-X Annotations.

While we strive to utilize as many datasets as possible in our study, we find that there are only a few datasets with neutral SMPL-X annotations and many datasets with female/male (gendered) SMPL-X annotations or SMPL annotations. An intuitive solution is to use the official fitting tool [35], however, this optimization-based tool is relatively slow to convert a large number of annotations (fitting takes $241 \pm 126$ seconds per frame). Hence, we experiment with a new approach.

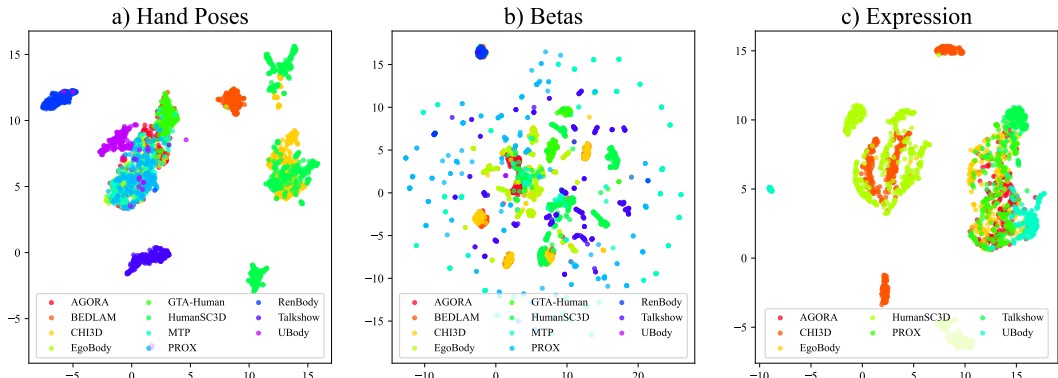

Figure 9: **Comparisons of hand pose, shape beta, and facial expression parameters distributions among different datasets**. We illustrate these distributions with UMAP [28]. The two axes are the two dimensions of the embedded space and have no unit.

For gendered SMPL-X annotations, we train a small adapter network $A$ (consisting of three layers of fully-connected layers) that takes in gendered body shape parameters $\beta_{f/m}$ and converts it neutral body shape parameters such that the following loss is minimized:

$$\mathcal{L} = ||M_{f/m}(\theta, \beta_{f/m}) - M_n(\theta, A(\beta_{f/m}))||_2 \tag{1}$$

where $M_{f/m}$ are gendered SMPL-X body model, and $M_n$ is the neutral SMPL-X body model, $\theta$ is body pose is obtained by random sampling in the embedding space of VPoser [35]. We test our adapter on AGORA [34] and find that the vertex-to-vertex error between ground truth gendered SMPL-X mesh and neutral SMPL-X with adapted neutral $\beta$ is 8.4 mm, which we consider to be sufficiently small. This approach is very fast (0.09 seconds per frame). Hence, we apply our adapter on AGORA, EgoBody, DNA-Rendering, and RICH.

However, we empirically find that the adapter does not work well across significantly different topologies (*i.e.* SMPL and SMPL-X), training similar adapters results in a 27.1 mm vertex-to-vertex error. Hence, for datasets with SMPL annotations, we only supervise ground truth global orientation and body pose. Although this is a slight abuse of the parameters (SMPL and SMPL-X parameters are not directly transferable), we find in our experiments that such a strategy leads to performance gains.

## D Additional Experiments and Analyses

### D.1 More Distribution Comparisons

In Fig. 9, we plot more distributions of additional parameters: a) hand poses, b) betas (body shape), and c) facial expression, all via UMAP dimension reduction. Datasets without proper SMPL-X parameters (*e.g.*, SMPL annotation only, or pseudo-annotated that typically have invalid hand poses) are not included in the study. For hand poses, we concatenate both left and right-hand parameters in rotation matrix representation. For betas and expression, we directly use their first 10 components. It is observed that datasets such as DNA-Rendering, CHI3D, HumanSC3D, and Talkshow form distinct clusters for hand poses and betas, and it is difficult to find any dataset to provide a well-spread coverage. For expression, there is still a lack of diverse datasets.

### D.2 Training Schemes

As shown in Table 1, we perform the ablation study for the training scheme. We investigate the effect of dataset selection. We selecte the bottom 5 and bottom 10 datasets according to our individual dataset benchmark rankings and trained the SMPLer-X-B model with the same number of instances as used in training with the top 5 and top 10 datasets.

It is proved that our training scheme is efficient. Selecting the top 5 or top 10 datasets according to the single dataset benchmark leads to a much better performance compared to selecting the bottom 5 or bottom 10 datasets. The foundation model can benefit from adding higher-ranked (i.e., Top 5/10)

Table 1: **Training schemes**. We study the different training schemes by comparing the model trained with the Top 5 / Top 10 datasets with the Bottom 5 / Bottom 10 datasets according to our individual dataset benchmark rankings.

| Method | Dataset | #Instance | MPE(mm) |
|---|---|---|---|
| SMPLer-X-B | Top5 | 0.75M | 103.47 |
| SMPLer-X-B | Bottom5 | 0.75M | 115.61 |
| SMPLer-X-B | Top10 | 1.5M | 89.20 |
| SMPLer-X-B | Bottom10 | 1.5M | 115.10 |

Table 2: **Finetuning strategies**. We study the different finetuning strategies by freezing the parameters in different parts of the network. Models are tested on UBody test set, and † denotes the models that are finetuned on UBody train set.

| Method | Finetune | #Param. | PA-PVE (*mm*) | | | PVE (*mm*) | | |
|---|---|---|---|---|---|---|---|---|
| | | | All | Hands | Face | All | Hands | Face |
| SMPLer-X-H32 | - | 662M | 29.9 | 9.8 | 2.6 | 54.5 | 36.4 | 20.6 |
| SMPLer-X-H32† | Full network | 662M | 27.8 | 9.0 | 2.3 | 51.3 | 32.6 | 19.1 |
| SMPLer-X-H32† | Neck+Head | 31M | 27.8 | 9.0 | 2.3 | 51.1 | 32.5 | 19.1 |
| SMPLer-X-H32† | Head | 5M | 29.9 | 9.7 | 2.6 | 54.2 | 35.9 | 20.6 |

data into training, while lower-ranked data (i.e., Bottom 5/10) is not as effective in improving the model's performance. Despite this, we finetune the entire network in all other finetuning experiments.

### D.3 Finetuning Strategies

In Table 2, we evaluate different strategies that finetune different parts of our foundation model. We observe that finetuning only the neck and head is very efficient: it achieves even slightly better performance than finetuning the entire network, with much fewer learnable parameters. We speculate that after training with a large number of datasets, the backbone is already very strong and generalizable. Hence, finetuning the backbone does not yield much performance improvement.

### D.4 Data Sampling Strategies

As for the sampling strategy, we did the ablation study on three different strategies, including 1) Balanced: we set all the datasets to have the same length; 2) Weighted: we set the dataset length according to the individual dataset benchmark rankings. Specifically, we sort the datasets based on their rankings and then assign weights to each dataset. As a result of this weighting, the datasets are upsampled or downsampled so that the lengths of the datasets are adjusted to an arithmetic sequence. The length of the dataset with the highest ranking is 4 times that of the dataset with the lowest ranking, and the sum of the total lengths of all datasets is fixed; 3) Concat: we simply concatenate all the datasets with their original length.

The performance of the foundation model is not sensitive to the sampling strategy as shown in Table 3, while the balanced strategy is more intuitive, easy to implement, and efficient, the weighted strategy may have more potential with more effort in weight tuning.

### D.5 Training Domains

In Table 4, we further study the impact of training domains. It is clear that in-domain training (including the training split of a dataset in the training, and testing on the test split of the same dataset) is highly effective, as "seeing" the dataset always brings significant performance improvement. However, we highlight that having out-of-domain training sets in the training is also highly effective: with 4 seen datasets fixed, SMPLer-X benefits tremendously from having 10, 20, and 32 datasets in training in terms of MPE. It is worth noting that training with a lot of datasets especially benefits out-of-domain ("unseen" benchmarks) performance as errors on EHF, ARCTIC, and DNA-Rendering-HiRes

Table 3: **Data sampling strategies**. We trained SMPLer-X-H32 models with different data sampling strategies.

| Method | Strategy | #Instance | MPE(mm) |
|---|---|---|---|
| SMPLer-X-H32 | balanced | 4.5M | 63.08 |
| SMPLer-X-H32 | weighted | 4.5M | 62.12 |
| SMPLer-X-H32 | concat | 5.6M | 63.32 |

Table 4: **Impact of training domains.** We investigate the impact of seeing the train split of a benchmark dataset during training and how this may affect the generalizability of a model. MPE: mean primary error of AGORA-val, EgoBody-EgoSet, UBody, 3DPW, and EHF. The yellow shaded numbers denote that the corresponding train split is used in training. Top-1 values are bolded, and the second best values are underlined. Except for 3DPW using MPJPE as the metric, other datasets are evaluated via PVE. Unit: mm. #Data.: number of datasets used in the training. #Seen: number of evaluation benchmarks' used in the training, note here that only benchmarks that are included in the MPE computation are counted, thus excluding ARCTIC and DNA-Rendering-HiRes. *: not following the standard dataset selection scheme.

| #Data. | #Seen | Model | MPE | AGORA [34] | EgoBody [40] | UBody [26] | 3DPW [37] | EHF [36] | ARCTIC [12] | DNA-R-HiRes [9] |
|---|---|---|---|---|---|---|---|---|---|---|
| 5 | 1 | SMPLer-X-L5 | 100.8 | 89.1 | 101.6 | 114.0 | 102.8 | 96.7 | 99.8 | 90.6 |
| 5 | 4 | SMPLer-X-L* | 85.2 | 96.7 | 81.9 | 68.1 | 95.5 | 83.6 | 103.6 | 98.2 |
| 10 | 2 | SMPLer-X-L10 | 80.6 | 82.6 | 69.7 | 104.0 | 82.5 | 64.0 | 76.9 | 76.2 |
| 10 | 4 | SMPLer-X-L* | 72.7 | 84.0 | 71.6 | 62.8 | 81.7 | 63.4 | 80.8 | 75.8 |
| 20 | 4 | SMPLer-X-L20 | 70.5 | 80.7 | 66.6 | 61.5 | 78.3 | 65.4 | 52.2 | 77.7 |
| 32 | 4 | SMPLer-X-L32 | **66.2** | **74.2** | **62.2** | **57.3** | **75.2** | 62.3 | **48.6** | **54.4** |

decrease with more datasets in the training set. Lastly, training on 32 datasets with our SMPLer-X-L obtains the best performance with 66.2 mm MPE, making it a strong and effective SMPL-X estimator.

# E   Complete Results

## E.1   Benchmarking EHPS Datasets on Training Sets

In the main paper, we benchmark individual datasets on the testing sets of the key EHPS evaluation benchmarks. However, this dataset benchmark is unsuitable for selecting top datasets for training EHPS, as the ranking leaks information about the testing sets to some extent. Hence, we construct a new benchmark that ranks EHPS datasets on the training set of AGORA, UBody, EgoBody, and 3DPW (EHF is omitted as it does not have a training set) in Table 12.

## E.2   Complete Results of Foundation Models on Evaluation Benchmarks

We show complete results including our strongest foundation model SMPLer-X-H32 on AGORA validation set (Table 5), UBody (Table 7), EgoBody-EgoSet (Table 8), EHF (Table 6), ARCTIC (Table 9) and DNA-Rendering-HiRes (Table 10).

Table 5: **AGORA Val set**. † denotes methods that are finetuned on the AGORA training set.

| Method | PA-PVE↓ (mm) | | | PVE↓ (mm) | | |
|---|---|---|---|---|---|---|
| | All | Hands | Face | All | Hands | Face |
| Hand4Whole [31]† | 73.2 | 9.7 | 4.7 | 183.9 | 72.8 | 81.6 |
| OSX [26]* | 45.0 | 8.5 | 3.9 | 79.6 | 48.2 | 37.9 |
| SMPLer-X-S5 | 72.1 | 10.2 | 5.1 | 119.0 | 66.8 | 58.9 |
| SMPLer-X-S10 | 67.5 | 10.2 | 4.8 | 116.0 | 65.2 | 57.5 |
| SMPLer-X-S20 | 62.1 | 10.0 | 4.4 | 109.2 | 63.3 | 55.2 |
| SMPLer-X-S32 | 58.7 | 9.8 | 4.2 | 105.2 | 61.9 | 53.9 |
| SMPLer-X-B5 | 63.8 | 9.6 | 4.8 | 102.7 | 59.0 | 50.8 |
| SMPLer-X-B10 | 58.4 | 9.5 | 4.6 | 97.8 | 57.8 | 49.1 |
| SMPLer-X-B20 | 56.9 | 9.3 | 4.3 | 95.6 | 56.5 | 47.9 |
| SMPLer-X-B32 | 52.0 | 9.2 | 4.1 | 88.0 | 54.5 | 45.9 |
| SMPLer-X-L5 | 56.1 | 9.2 | 4.3 | 88.3 | 53.0 | 43.3 |
| SMPLer-X-L10 | 50.6 | 9.1 | 4.1 | 82.6 | 51.9 | 42.3 |
| SMPLer-X-L20 | 48.6 | 8.9 | 4.0 | 80.7 | 51.0 | 41.3 |
| SMPLer-X-L32 | 45.1 | 8.7 | 3.8 | 74.2 | 47.8 | 38.7 |
| SMPLer-X-H5 | 57.8 | 9.1 | 4.2 | 89.0 | 52.6 | 42.6 |
| SMPLer-X-H10 | 51.0 | 9.0 | 4.0 | 81.4 | 51.4 | 40.5 |
| SMPLer-X-H20 | 47.1 | 8.8 | 3.9 | 77.5 | 49.5 | 39.4 |
| SMPLer-X-H32 | 42.9 | 8.5 | 3.7 | 69.5 | 45.6 | 35.9 |
| SMPLer-X-H32† | 41.0 | 8.2 | 3.7 | 65.4 | 43.8 | 34.0 |

Table 6: **EHF**. As EHF does not have a training set, we do not perform finetuning.

| Method | PA-PVE↓ (mm) | | | PVE↓ (mm) | | |
|---|---|---|---|---|---|---|
| | All | Hands | Face | All | Hands | Face |
| Hand4Whole [31] | 50.3 | 10.8 | 5.8 | 76.8 | 39.8 | 26.1 |
| OSX [26] | 48.7 | 15.9 | 6.0 | 70.8 | 53.7 | 26.4 |
| SMPLer-X-S5 | 70.7 | 16.0 | 5.9 | 100.5 | 64.0 | 27.1 |
| SMPLer-X-S10 | 60.5 | 16.0 | 5.7 | 89.9 | 59.1 | 22.3 |
| SMPLer-X-S20 | 51.0 | 15.5 | 5.5 | 86.6 | 54.7 | 22.1 |
| SMPLer-X-S32 | 50.5 | 14.8 | 5.2 | 74.1 | 54.6 | 20.0 |
| SMPLer-X-B5 | 61.4 | 15.4 | 5.8 | 96.1 | 58.4 | 27.1 |
| SMPLer-X-B10 | 46.7 | 15.7 | 5.6 | 74.7 | 55.1 | 21.3 |
| SMPLer-X-B20 | 41.9 | 15.9 | 5.3 | 73.0 | 53.7 | 20.8 |
| SMPLer-X-B32 | 40.7 | 14.5 | 5.2 | 67.3 | 52.1 | 20.6 |
| SMPLer-X-L5 | 53.9 | 14.7 | 5.9 | 89.5 | 57.8 | 29.9 |
| SMPLer-X-L10 | 40.7 | 15.6 | 5.3 | 64.0 | 52.9 | 18.1 |
| SMPLer-X-L20 | 37.8 | 15.0 | 5.1 | 65.4 | 49.4 | 17.4 |
| SMPLer-X-L32 | 37.1 | 14.1 | 5.0 | 62.4 | 47.1 | 17.0 |
| SMPLer-X-H5 | 47.0 | 14.3 | 5.9 | 68.3 | 55.6 | 25.0 |
| SMPLer-X-H10 | 40.1 | 15.6 | 5.2 | 56.6 | 50.2 | 18.9 |
| SMPLer-X-H20 | 39.0 | 14.4 | 5.0 | 59.4 | 47.1 | 17.8 |
| SMPLer-X-H32 | 39.0 | 14.8 | 5.0 | 56.8 | 42.2 | 19.0 |

Table 7: **UBody.** † denotes the methods that are finetuned on the UBody training set.

| Method | PA-PVE↓ (mm) | | | PVE↓ (mm) | | |
|---|---|---|---|---|---|---|
| | All | Hands | Face | All | Hands | Face |
| Hand4Whole [31] | 42.2 | 8.3 | 3.1 | 95.7 | 39.0 | 31.2 |
| OSX [26]† | 42.2 | 8.6 | 2.0 | 81.9 | 41.5 | 21.2 |
| SMPLer-X-S5 | 53.9 | 11.8 | 3.9 | 110.1 | 59.4 | 34.5 |
| SMPLer-X-S10 | 50.4 | 11.5 | 3.7 | 107.7 | 57.4 | 32.8 |
| SMPLer-X-S20 | 37.5 | 11.1 | 3.2 | 70.7 | 49.6 | 26.1 |
| SMPLer-X-S32 | 36.4 | 10.7 | 3.0 | 68.1 | 47.8 | 25.0 |
| SMPLer-X-B5 | 52.3 | 11.9 | 3.8 | 105.8 | 56.9 | 32.6 |
| SMPLer-X-B10 | 49.7 | 12.0 | 3.6 | 107.3 | 57.1 | 31.7 |
| SMPLer-X-B20 | 35.5 | 11.0 | 3.0 | 65.3 | 46.9 | 23.4 |
| SMPLer-X-B32 | 33.7 | 10.8 | 2.8 | 63.3 | 43.9 | 22.7 |
| SMPLer-X-L5 | 51.8 | 12.5 | 3.6 | 110.8 | 56.3 | 37.5 |
| SMPLer-X-L10 | 48.0 | 12.8 | 3.5 | 104.0 | 56.1 | 32.0 |
| SMPLer-X-L20 | 33.2 | 10.6 | 2.8 | 61.5 | 43.3 | 23.1 |
| SMPLer-X-L32 | 30.9 | 10.2 | 2.7 | 57.3 | 39.2 | 21.6 |
| SMPLer-X-H5 | 48.1 | 12.1 | 3.7 | 102.1 | 53.3 | 33.4 |
| SMPLer-X-H10 | 48.5 | 12.6 | 3.5 | 100.7 | 54.8 | 30.9 |
| SMPLer-X-H20 | 32.8 | 10.3 | 2.8 | 59.9 | 41.0 | 22.7 |
| SMPLer-X-H32 | 29.9 | 9.8 | 2.6 | 54.5 | 36.4 | 20.6 |
| SMPLer-X-H32† | 27.8 | 9.0 | 2.3 | 51.3 | 32.6 | 19.1 |

Table 8: **EgoBody-EgoSet.** † are finetuned on the EgoBody-EgoSet training set.

| Method | PA-PVE↓ (mm) | | | PVE↓ (mm) | | |
|---|---|---|---|---|---|---|
| | All | Hands | Face | All | Hands | Face |
| Hand4Whole [31] | 58.8 | 9.7 | 3.7 | 121.9 | 50.0 | 42.5 |
| OSX [26]† | 45.3 | 10.0 | 3.0 | 82.3 | 46.8 | 35.2 |
| SMPLer-X-S5 | 62.8 | 10.8 | 4.1 | 114.2 | 53.3 | 44.3 |
| SMPLer-X-S10 | 52.2 | 10.0 | 3.4 | 88.6 | 48.6 | 37.6 |
| SMPLer-X-S20 | 48.1 | 10.0 | 3.3 | 84.3 | 47.2 | 37.8 |
| SMPLer-X-S32 | 46.0 | 10.0 | 3.1 | 82.5 | 46.0 | 36.2 |
| SMPLer-X-B5 | 59.4 | 10.6 | 4.0 | 108.1 | 48.0 | 40.0 |
| SMPLer-X-B10 | 45.3 | 10.1 | 3.2 | 76.4 | 45.5 | 32.4 |
| SMPLer-X-B20 | 43.8 | 9.9 | 3.2 | 75.5 | 44.6 | 32.7 |
| SMPLer-X-B32 | 40.7 | 9.9 | 3.1 | 72.7 | 43.7 | 32.4 |
| SMPLer-X-L5 | 52.9 | 10.5 | 3.8 | 98.7 | 45.2 | 39.1 |
| SMPLer-X-L10 | 40.5 | 10.0 | 3.0 | 69.7 | 43.1 | 32.0 |
| SMPLer-X-L20 | 38.9 | 9.9 | 3.0 | 66.6 | 42.7 | 31.8 |
| SMPLer-X-L32 | 36.3 | 9.8 | 2.9 | 62.2 | 41.4 | 30.7 |
| SMPLer-X-H5 | 48.0 | 10.5 | 3.4 | 87.4 | 43.5 | 37.5 |
| SMPLer-X-H10 | 38.8 | 10.0 | 2.9 | 65.7 | 42.6 | 31.1 |
| SMPLer-X-H20 | 36.7 | 9.8 | 2.9 | 63.5 | 41.3 | 30.8 |
| SMPLer-X-H32 | 34.3 | 9.8 | 2.7 | 59.5 | 39.6 | 28.7 |
| SMPLer-X-H32† | 33.9 | 10.0 | 2.5 | 57.0 | 40.2 | 27.1 |

Table 9: **ARCTIC**. † denotes the methods that are finetuned on the ARCTIC training set.

| Method | PA-PVE↓ (mm) | | | PVE↓ (mm) | | |
|---|---|---|---|---|---|---|
| | All | Hands | Face | All | Hands | Face |
| Hand4Whole [31] | 63.4 | 18.1 | 4.0 | 136.8 | 54.8 | 59.2 |
| OSX [26]† | 33.0 | 18.8 | 3.3 | 58.4 | 39.4 | 30.4 |
| SMPLer-X-S5 | 66.1 | 16.7 | 4.0 | 117.3 | 58.7 | 46.5 |
| SMPLer-X-S10 | 58.8 | 17.5 | 3.2 | 104.6 | 56.6 | 41.1 |
| SMPLer-X-S20 | 37.6 | 18.9 | 2.7 | 58.7 | 45.2 | 30.5 |
| SMPLer-X-S32 | 34.5 | 18.9 | 2.7 | 55.3 | 42.9 | 28.9 |
| SMPLer-X-B5 | 66.3 | 16.9 | 3.4 | 105.4 | 55.6 | 41.4 |
| SMPLer-X-B10 | 54.0 | 17.9 | 2.5 | 85.2 | 53.4 | 35.0 |
| SMPLer-X-B20 | 34.9 | 18.9 | 2.7 | 56.3 | 40.9 | 29.6 |
| SMPLer-X-B32 | 31.9 | 19.0 | 2.8 | 52.6 | 40.1 | 27.4 |
| SMPLer-X-L5 | 57.2 | 17.0 | 2.9 | 95.1 | 52.8 | 37.7 |
| SMPLer-X-L10 | 46.9 | 18.1 | 2.3 | 76.9 | 50.8 | 33.2 |
| SMPLer-X-L20 | 31.9 | 18.9 | 2.5 | 52.2 | 39.3 | 27.0 |
| SMPLer-X-L32 | 29.4 | 18.9 | 2.7 | 48.6 | 38.8 | 26.8 |
| SMPLer-X-H5 | 49.3 | 17.4 | 2.5 | 79.9 | 49.3 | 33.9 |
| SMPLer-X-H10 | 41.4 | 18.8 | 2.1 | 71.6 | 49.3 | 30.9 |
| SMPLer-X-H20 | 29.3 | 18.9 | 2.5 | 48.5 | 38.3 | 26.3 |
| SMPLer-X-H32 | 27.6 | 18.7 | 2.6 | 44.6 | 36.9 | 24.6 |
| SMPLer-X-H32† | 27.7 | 18.8 | 2.6 | 44.7 | 37.0 | 24.7 |

Table 10: **DNA-Rendering-HiRes**. † are finetuned on the DNA-Rendering-HiRes training set.

| Method | PA-PVE↓ (mm) | | | PVE↓ (mm) | | |
|---|---|---|---|---|---|---|
| | All | Hands | Face | All | Hands | Face |
| Hand4Whole [31] | 62.8 | 11.0 | 4.2 | 111.4 | 56.4 | 52.6 |
| OSX [26]† | 43.5 | 7.5 | 3.5 | 67.1 | 43.3 | 38.2 |
| SMPLer-X-S5 | 70.9 | 10.4 | 4.7 | 104.9 | 57.6 | 49.7 |
| SMPLer-X-S10 | 63.9 | 11.0 | 4.4 | 98.4 | 57.0 | 47.3 |
| SMPLer-X-S20 | 55.6 | 10.2 | 4.4 | 87.3 | 53.3 | 46.2 |
| SMPLer-X-S32 | 47.1 | 7.7 | 3.5 | 70.1 | 46.9 | 39.0 |
| SMPLer-X-B5 | 59.9 | 10.5 | 4.3 | 91.1 | 50.5 | 44.6 |
| SMPLer-X-B10 | 53.3 | 11.5 | 4.1 | 83.7 | 50.5 | 42.4 |
| SMPLer-X-B20 | 50.7 | 11.7 | 4.2 | 83.3 | 50.9 | 43.5 |
| SMPLer-X-B32 | 40.9 | 7.4 | 3.4 | 61.9 | 40.5 | 36.6 |
| SMPLer-X-L5 | 52.4 | 10.3 | 4.0 | 85.9 | 47.6 | 44.5 |
| SMPLer-X-L10 | 47.0 | 11.2 | 3.8 | 76.2 | 47.8 | 41.7 |
| SMPLer-X-L20 | 44.4 | 11.1 | 4.5 | 77.7 | 47.5 | 43.2 |
| SMPLer-X-L32 | 35.8 | 7.2 | 3.2 | 54.4 | 36.7 | 34.0 |
| SMPLer-X-H5 | 53.9 | 10.3 | 3.9 | 81.9 | 46.3 | 40.7 |
| SMPLer-X-H10 | 47.4 | 10.9 | 3.7 | 76.2 | 47.0 | 39.0 |
| SMPLer-X-H20 | 43.0 | 11.2 | 3.8 | 72.8 | 45.6 | 40.5 |
| SMPLer-X-H32 | 34.0 | 7.1 | 3.1 | 51.4 | 34.5 | 32.0 |
| SMPLer-X-H32† | 32.7 | 7.1 | 3.1 | 49.8 | 33.2 | 30.8 |

Table 11: **3DPW.** †denotes the models that are finetuned on the 3DPW training set. Only whole-body (SMPL-X) methods are listed. Unit: *mm*.

| Method | MPJPE | PA-MPJPE |
|---|---|---|
| Hand4Whole [31] | 86.6 | 54.4 |
| OSX [26]† | 86.2 | 60.6 |
| SMPLer-X-S5 | 110.2 | 79.1 |
| SMPLer-X-S10 | 97.4 | 69.0 |
| SMPLer-X-S20 | 87.4 | 60.0 |
| SMPLer-X-S32 | 83.2 | 57.1 |
| SMPLer-X-B5 | 104.8 | 72.0 |
| SMPLer-X-B10 | 89.9 | 62.7 |
| SMPLer-X-B20 | 83.5 | 57.6 |
| SMPLer-X-B32 | 80.3 | 53.4 |
| SMPLer-X-L5 | 97.8 | 62.6 |
| SMPLer-X-L10 | 82.5 | 56.0 |
| SMPLer-X-L20 | 78.3 | 52.1 |
| SMPLer-X-L32 | 75.2 | 50.5 |
| SMPLer-X-H5 | 88.3 | 60.3 |
| SMPLer-X-H10 | 78.7 | 54.8 |
| SMPLer-X-H20 | 74.4 | 50.9 |
| SMPLer-X-H32 | 75.0 | 50.6 |
| SMPLer-X-H32† | **71.7** | **48.0** |

Table 12: **Selection of training datasets by ranking on the training set of key benchmarks.** For each dataset, we evaluate a model trained on the training set and on the *training* sets of four major benchmarks: AGORA, UBody, EgoBody (EgoSet), and 3DPW. Datasets are then ranked by MPE. ★: ranking on MPE. Top 1 values on each benchmark are bolded, and the rest of Top-5 are underlined.

| Dataset | MPE↓ | AGORA [34]↓ | UBody [26]↓ | EgoBody [40]↓ | 3DPW [37]↓ |
|---|---|---|---|---|---|
| BEDLAM [5] | **124.7** | 167.8 | 126.7 | 106.3 | **98.1** |
| AGORA [34] | 129.9 | **131.7** | 124.4 | 134.2 | 131.2 |
| GTA-Human [8] | 135.1 | 164.2 | 137.6 | 135.2 | 103.5 |
| SynBody [38] | 138.6 | 172.3 | 146.0 | 129.7 | 106.3 |
| InstaVariety [22] | 139.6 | 198.2 | 128.4 | 131.6 | 100.6 |
| MSCOCO [27] | 139.7 | 196.8 | 110.4 | 130.5 | 121.1 |
| SPEC [23] | 150.0 | 166.2 | 138.8 | 155.4 | 139.7 |
| EgoBody-MVSet [40] | 151.8 | 193.3 | 194.7 | 119.7 | 99.3 |
| MPII [2] | 152.0 | 205.5 | 127.3 | 143.3 | 131.9 |
| RICH [17] | 155.7 | 198.9 | 171.8 | 136.9 | 115.2 |
| Egobody-EgoSet [40] | 157.1 | 213.6 | 123.5 | **63.6** | 134.1 |
| CrowdPose [25] | 162.3 | 213.0 | 133.7 | 146.2 | 156.3 |
| MuCo-3DHP [30] | 163.4 | 193.2 | 189.7 | 151.1 | 119.7 |
| UBody [26] | 166.6 | 212.9 | **61.5** | 137.6 | 149.2 |
| PROX [16] | 167.3 | 205.1 | 186.8 | 145.2 | 132.1 |
| MPI-INF-3DHP [29] | 167.5 | 221.3 | 167.4 | 150.0 | 131.4 |
| PoseTrack [1] | 177.0 | 219.2 | 165.4 | 173.2 | 150.2 |
| BEHAVE [4] | 179.0 | 204.8 | 212.3 | 167.2 | 131.8 |
| HumanSC3D [14] | 184.8 | 213.8 | 237.7 | 174.8 | 112.9 |
| CHI3D [13] | 192.3 | 209.2 | 256.7 | 180.7 | 122.5 |
| Human3.6M [18] | 207.4 | 224.5 | 282.4 | 210.7 | 112.1 |
| DNA-R.-HiRes [9] | 207.5 | 231.1 | 275.4 | 189.4 | 134.0 |
| ARCTIC [12] | 222.5 | 303.6 | 205.9 | 177.3 | 203.2 |
| Talkshow [39] | 225.3 | 290.0 | 132.2 | 188.1 | 290.8 |
| UP3D [24] | 226.0 | 257.4 | 226.8 | 208.4 | 211.6 |
| 3DPW [37] | 230.6 | 231.3 | 266.0 | 194.5 | 140.6 |
| DNA-Rendering [9] | 253.2 | 288.7 | 342.5 | 234.4 | 147.2 |
| MTP [33] | 270.5 | 272.8 | 284.8 | 259.2 | 265.4 |
| FIT3D [15] | 272.9 | 323.5 | 392.8 | 242.7 | 132.5 |
| OCHuman [41] | 282.3 | 307.7 | 266.7 | 261.5 | 293.4 |
| LSPET [20] | 330.2 | 361.6 | 301.8 | 317.3 | 340.2 |
| SSP3D [36] | 512.0 | 545.9 | 533.4 | 529.7 | 439.1 |