# OpenReview forum: "SMPLer-X: Scaling Up Expressive Human Pose and Shape Estimation"
_NeurIPS.cc/2023/Track/Datasets_and_Benchmarks — NeurIPS 2023 Datasets and Benchmarks Poster_

### Official Review · Reviewer_5yuA · 2023-07-20
**This paper integrates 4.5 million instances from 32 pose estimation datasets and proposed a universal model called SMPLer-X. However, it lacks appropriate motivation and innovation.**

**Rating:** 4
**Confidence:** 5
**Correctness:** Yes
**Clarity:** Well written.

**Strengths:**

1. The author's design of SMPLer-X demonstrates the immense potential of large models and highlights the significance of combining large models with large datasets.
2. The author extensively researched various datasets and conducted detailed analyses of their attributes.
3. The abundant experimental evidence presented in the supplementary materials validates the comprehensiveness of the experiments.


**Additional Feedback:**

The authors need to think further about what is the point of integrating 32 datasets. How to better utilize the existing datasets?

**Documentation:**

Reproducible Work

**Limitations:**

The authors have discussed this briefly in their paper.

**Opportunities For Improvement:**

In general, this article is a good technical report where the author analyzes existing 3D datasets and designs a large-scale pose estimation model based on ViT. However, the article lacks appropriate motivation and innovation.

Firstly, although the author performs statistical analysis on existing datasets, no further work is done, such as merging these datasets in a certain way and eliminating redundant data that may affect performance (as indicated in the analysis based on Fig. 3a). This greatly diminishes the novelty of the paper.

Secondly, the proposed SMPLer-X model is merely an incremental work based on ViT and cannot be considered the primary innovation of the paper.

Lastly, I believe the author should discuss the model's runtime speed (e.g., frames per second) to consider the real-time aspect of the pose estimation network.


**Relation To Prior Work:**

Yes.

**Summary And Contributions:**

To address the issue of limited dataset size in 3D pose estimation, the authors integrated 4.5 million instances from 32 pose estimation datasets and proposed a universal model called SMPLer-X. This model leverages big data to achieve state-of-the-art performance on various benchmark tests. Additionally, the authors conducted a detailed analysis of attribute distributions across different datasets.

---

> ### Author Response · Authors · 2023-08-22
> **Author Response (1/2)**
>
> We would like to thank the reviewer for the comments. Below we hope to explain point by point.
>
> ---
>
> > **Q1**: The article lacks appropriate motivation and innovation.
>
> **A1**: We wish to emphasize our primary motivation: despite the recent advancements in EHPS (Expressive Human Pose and Shape) algorithms, contemporary state-of-the-art methods (SoTAs) consistently fall short in delivering robust results across a spectrum of scenarios. A critical observation is that many of these methods are trained predominantly on a limited number of datasets. This is particularly concerning given the surge in the availability of comprehensive, novel datasets. In terms of innovation, our research stands out as the first comprehensive investigation in scaling up both training data and model size of EHPS models. As a testament to our new insights, our foundational models outperform previous SoTA by clear margins across various benchmarks.
>
> ---
>
> > **Q2**: Firstly, although the author performs statistical analysis on existing datasets, no further work is done, such as merging these datasets in a certain way and eliminating redundant data that may affect performance (as indicated in the analysis based on Fig. 3a). This greatly diminishes the novelty of the paper.
>
> **A2**: Our primary innovation is that we are the pioneering study to scale training data and model size for EHPS, achieving unprecedented results as showcased in Table 3 of the main paper (SoTA on seven benchmarks). Moreover, the statistical analysis is in fact one of our key contributions: our efforts in the comprehensive analysis of the strengths and generalization abilities of 32 existing datasets and their key attributes provide new information about training strong EHPS models. Moreover, training foundation models using a wide variety of datasets presents complex challenges. To address these, we introduce novel techniques detailed in Supplementary Material Sec. C.3, which facilitates bridging discrepancies in data conventions.
>
> Subsequently, we present an empirical analysis focusing on data merging strategies and redundancy.
>
> | Model | Sampling strategy | #Instance | MPE(mm)|
> | --- | --- | --- |--- |
> | SMPLer-X-H32 | balanced | 4.5M | 63.08 |
> | SMPLer-X-H32 | weighted | 4.5M | 62.12 |
> | SMPLer-X-H32 | concat | 5.6M | 63.32 |
> Here we evaluate three strategies:
> - Balanced: we set all the datasets to have same length as mentioned in the paper (L254-L257).
> - Weighted: we set the dataset length according to the benchmark (Table 1) rankings. Specifically, we sort the datasets based on their rankings and then assign weights to each dataset. As a result of this weighting, the datasets are upsampled or downsampled so that the lengths of the datasets are adjusted to an arithmetic sequence. The length of the dataset with the highest ranking is 4 times that of the dataset with the lowest ranking, and the sum of the total lengths of all datasets is fixed.
> - Concat: we simply concatenate all the datasets with their original length.
>
> We observe that the foundation model performance is not sensitive to the merging and sampling strategy. Particularly, changing strategy only leads to <1 mm difference in MPE compared to our current *balanced* strategy.
>
> Moreover, for data redundancy, we highlight that: 1) By comparing *balanced* with *concat* strategies, we find that our *balanced* uses fewer training instances but achieves slightly better results. This indicates that there is indeed some data redundancy among all 5.6M data. 2) By comparing *balanced* with *weighted* strategies, we observe that the *balanced* strategy achieves results close to *weighted*, despite the fact that there is less redundancy as higher-ranked datasets are prioritized in the *weighted* training. Interestingly, the difference (<1 mm) is clearly not as much as scaling up the training instance (e.g., >30 mm improvements in MPE from SMPLer-X-L5 to SMPLer-X-L32).
>
> In summary, our experiments demonstrate that increasing the number of training instances yields more substantial performance improvements than refining merging strategies or eliminating data redundancy. These latter areas might be explored further in future research.

---

> > ### Author Response · Authors · 2023-08-22
> > **Author Response (2/2)**
> >
> > > **Q3**: the proposed SMPLer-X model is merely an incremental work based on ViT and cannot be considered the primary innovation of the paper.
> >
> > **A3**: Thank you for your comment. We would like to clarify that:
> > - This work focuses on benchmarking existing datasets for EHPS, from which we obtain significant insights (Sec. 3) and train strong foundation models. Particularly, we have stated in Line 248 that “the design of SMPLer-X is not the focus of our investigation”.
> > - Non-algorithmic works can also contribute significantly to the research community. We refer to the official [blogpost](https://neuripsconf.medium.com/announcing-the-neurips-2021-datasets-and-benchmarks-track-644e27c1e66c) for NeurIPS Dataset and Benchmark Track, in which we quote “the vast majority of the NeurIPS community focuses on algorithm design, but often can’t easily find good datasets …  many researchers resort to data that are conveniently available, but not representative of real applications … (This new track) will serve as a venue … for discussions on how to improve dataset development and data-oriented work more broadly”.
> > - Our work is aligned with the scope of the NeurIPS Dataset and Benchmark Track. According to the official instructions in “Call for Datasets and Benchmarks”, the scope of our work fits the following topics:
> >     - audits of existing datasets, identifying significant problems with existing datasets and their use
> >     - benchmarks on new or existing datasets
> > - Previous publications in this track often do not have algorithmic innovation as they benchmark existing methods and/or datasets to obtain new insights [A, B, C, D, E].
> >
> > [A] Maekawa et al., “Beyond Real-world Benchmark Datasets: An Empirical Study of Node Classification with GNNs”. NeurIPS 2022 Datasets and Benchmarks Track.
> >
> > [B] Pang et al., “Benchmarking and Analyzing 3D Human Pose and Shape Estimation Beyond Algorithms”. NeurIPS 2022 Datasets and Benchmarks Track.
> >
> > [C] Bao et al., “It’s COMPASlicated: The Messy Relationship between RAI Datasets and Algorithmic Fairness Benchmarks”. NeurIPS 2021 Datasets and Benchmarks Track.
> >
> > [D] Li et al. "MQBench: Towards Reproducible and Deployable Model Quantization Benchmark." NeurIPS 2021 Datasets and Benchmarks Track.
> >
> > [E] Yi et al. “Benchmarking the Robustness of Spatial-Temporal Models Against Corruptions.” NeurIPS 2021 Datasets and Benchmarks Track.
> >
> > ---
> >
> > > **Q4**：I believe the author should discuss the model's runtime speed (e.g., frames per second) to consider the real-time aspect of the pose estimation network.
> >
> > **A4**: We tested the inference speed of the SMPLer-X model family on a single V100 GPU, and the inference speed in FPS (frames per second) is shown below. SMPLer-X family is faster than OSX and the smaller versions (SMPLer-X-S/B) can achieve real-time performance, with SMPLer-X-L on the verge of achieving real-time speeds.
> > | Model | FPS |
> > | --- | --- |
> > | OSX | 12.17 |
> > | SMPLer-X-S | 36.17 |
> > | SMPLer-X-B | 33.09 |
> > | SMPLer-X-L | 24.44 |
> > | SMPLer-X-H | 17.47 |

---

> ### Author Response · Authors · 2023-08-28
> **Follow-up on Author Response**
>
> Dear Reviewer,
>
> We genuinely value and respect the detailed feedback you provided on our work. Recognizing the concerns raised, we have undertaken comprehensive experiments/analyses and provided a thorough response to each point. Your expertise has been paramount in directing our improvements, and we hope our previous response has adequately addressed your concerns. We would be truly grateful if you could reconsider our work in light of the clarifications and discussions that we have provided.
>
> Thank you for your time and dedication. We eagerly await your thoughts and are open to any further suggestions or clarifications you might have.

---

> > ### Comment · Reviewer_5yuA · 2023-08-30
> >
> > This paper studies the usage of multiple existing datasets, instead of presenting the new dataset. I am not sure whether this meets the requirements of this track. A similar paper was presented at ECCV 2022 DEMO TRACE: Accurate and Efficient Absolute 3D Human Pose Estimation Trained on Dozens of Datasets, which uses dozens of datasets for training 3D human pose estimation.
> > As a research paper, the technical contribution is relatively limited.  Based on these concerns, I keep my original rating.

---

> > > ### Author Response · Authors · 2023-08-30
> > > **Author Response Round 2**
> > >
> > > We appreciate the reviewer's feedback and would like to provide further clarifications.
> > >
> > > 1. The NeurIPS Dataset and Benchmarks Track caters not just to dataset papers but also to works that benchmark existing datasets. We kindly direct the reviewer to A3 in the Author Response, where we have quoted the official blogpost and guidelines for reference. In addition to [A-E], we highlight that similar works that develop benchmarks with existing datasets have been published in this track [F, G, H, I].
> > >
> > > 2. In Sec.2 Related Works, we have referenced the most recent version [44] of TRACE, and highlighted that this work focuses on human skeletons (keypoints), whereas our work studies the human parametric model that considers the estimation of both body pose and shape (meshes). These two directions have been established by the community as two distinct fields, with differences in baselines, training paradigms, metrics, and key benchmarks.
> > >
> > > 3. Regarding our technical contribution, we adhere to the principle that "simplicity is beauty". This philosophy is consistent with recognized works like DALL-E [J], GPT [K], and SAM [L], all of which have garnered significant acclaim within the community.
> > >
> > > 4. Our primary objective is to pioneer foundation models for EHPS. In this endeavor, gaining deep insights from the training data is critical. To this end, we have presented extensive experiments and analyses in Sec. 3. We believe these insights, particularly about scaling up EHPS, will be of immense value to the community.
> > >
> > > 5. Our understanding of big data and large models has empowered us to achieve notable performance on seven key benchmarks. Additionally, our models demonstrate a superior generalization capability, emphasizing their immense potential as foundational models in this domain.
> > >
> > > In light of these clarifications, we respectfully ask the reviewer to reconsider our submission. We appreciate the reviewer's time in evaluating our work and remain open to any further feedback or suggestions.
> > >
> > > [F] Kim et al., How Transferable are Video Representations Based on Synthetic Data?, NeurIPS 2022 Datasets and Benchmarks Track.
> > >
> > > [G] Wu et al., BackdoorBench: A Comprehensive Benchmark of Backdoor Learning, NeurIPS 2022 Datasets and Benchmarks Track.
> > >
> > > [H] Zhang et al., WRENCH: A Comprehensive Benchmark for Weak Supervision, NeurIPS 2021 Datasets and Benchmarks Track.
> > >
> > > [I] Wiegreffe et al., Teach Me to Explain: A Review of Datasets for Explainable Natural Language Processing, NeurIPS 2021 Datasets and Benchmarks Track.
> > >
> > > [J] Ramesh et al., Zero-Shot Text-to-Image Generation, ICML’21
> > >
> > > [K] Brown et al., Language Models are Few-Shot Learners, NeurIPS’20
> > >
> > > [L] Kirillov et al., Segment Anything, arXiv’23

---

### Official Review · Reviewer_uVnR · 2023-07-20
**borderline accept**

**Rating:** 6
**Confidence:** 4
**Correctness:** The method proposed is sound.

**Strengths:**

The dataset is quite large.
The performance obtained is clearly better than others.


**Additional Feedback:**

None

**Clarity:**

The quality of English is good; while I recommend authors to involve more examples.

**Documentation:**

Data description seems clear enough.

**Limitations:**

The paper proposed the dataset and it seemingly leads to better performance compared to others. However, there is no much visualization for the samples and it is hard to judge there exists diverse combination of body, hands and face motions.

**Opportunities For Improvement:**

Fig. 2 shows only body pose space, while not showing the hands and face motion space. I think authors could add plots for those two spaces.
Not much examples shown, it is hard to judge that the dataset indeed have diverse combination of human body, hands and face motions.

**Relation To Prior Work:**

Authors have novelty to propose the dataset having body, hands and face motions altogether. There is not much work for this stream of research.

**Summary And Contributions:**

The paper proposes the large-scale dataset for expressive human pose and shape reconstruction. It unifies body, hands and face motion captures. Authors also proposed the ViT-based baseline using the datasets. Experiments demonstrate the superiority of the proposed dataset compared to others.

---

> ### Author Response · Authors · 2023-08-22
> **Author Response**
>
> We would like to thank the reviewer for the comments.
>
> ---
>
> > **Q1**: Plots for hands and face space.
>
> **A1**: Thank you for your advice. We have included hand pose distribution in Supplementary Material Figure 10a), and the face expression distribution in this [Figure](https://github.com/caizhongang/SMPLer-X/blob/main/assets/expression_distribution.pdf). Note that we only illustrate datasets with hand and face parameters. It is observed that datasets indeed have vastly different distributions and a collective distribution (combined distribution of multiple datasets) would offer a larger coverage of the embedded space.
>
> ---
>
> > **Q2**: The paper proposed the dataset and it seemingly leads to better performance compared to others. However, there is no much visualization for the samples and it is hard to judge there exists diverse combination of body, hands and face motions.
>
> **A2**: Thank you for your suggestion. We encourage readers to refer to Supplementary Material Sec. B, where we visualize all datasets used in our study, with SMPL-X overlays. We highlight that these datasets have vastly different attributes including highly diverse body/hand poses and facial expressions. For example, ARCTIC [12] consists of comprehensive hand-object interaction motions, PoseTrack [1] consists of diverse human group activities, UBody [28] consists of a wide range of upper body movements with detailed hand gestures and facial expressions. The distribution analysis in the main paper, the Supplementary Material, and the rebuttal [Figure](https://github.com/caizhongang/SMPLer-X/blob/main/assets/expression_distribution.pdf) highlight the comprehensive coverage by integrating multiple datasets.
>
> Moreover, we would like to clarify that this study benchmarks 32 renowned datasets (instead of building a brand new one), making it the most extensive research on existing data for expressive human pose and shape estimation. Furthermore, our benchmarking methodology is designed to seamlessly accommodate additional datasets as they become available.

---

> ### Author Response · Authors · 2023-08-28
> **Follow-up on Author Response**
>
> Dear Reviewer,
>
> We are truly grateful for the insightful feedback you have shared regarding our work. Your constructive comments have guided us in refining our work. We have carefully addressed the comments in our previous response. If there are any more areas of concern or questions, we would be eager to address them.
>
> We would like to express our sincerest gratitude for your attention and consideration.

---

### Official Review · Reviewer_6mpM · 2023-07-21
**Review for SMPLer-X**

**Rating:** 7
**Confidence:** 4
**Correctness:** Yes, this is a benchmark submission.
**Clarity:** Yes.

**Strengths:**

1. Comprehensive analyses of existing 3D human mesh datasets, which provide the community with valuable tips for future dataset collection.

2. By benchmarking existing datasets, this paper provides several insights for training and finetuning a foundation model.

**Additional Feedback:**

No.

**Documentation:**

Yes.

**Ethics:**

Yes.

**Limitations:**

Yes.

**Opportunities For Improvement:**

1. Provide results of using the CNN-based foundation model which may give insights of designing CNN-based vs. ViT-based foundation models.

2. A few typos could be corrected.

**Relation To Prior Work:**

Yes.

**Summary And Contributions:**

This paper benchmarks 32 human pose and shape datasets to explore existing data resources comprehensively. They study the scaling-up issue by proposing a generalist foundation model, SMPLer-X. The paper provides analyses of the existing dataset attributes and benchmarks commonly used or newly proposed datasets. Several useful conclusions for future dataset collections are drawn and also provide insights for training and finetuning a foundation model in the 3D human mesh estimation field.

---

> ### Author Response · Authors · 2023-08-22
> **Author Response**
>
> We would like to thank the reviewer for the insightful reviews.
>
> ---
>
> > **Q1**: Provide results of using the CNN-based foundation model which may give insights of designing CNN-based vs. ViT-based foundation models.
>
> **A1**: We choose ViT architecture as our backbone since it has been a popular choice for building large models, and this is the first comprehensive investigation in scaling up both training data and model size of EHPS models.
> Though the transformer-based architecture cannot be trivially changed to a CNN-based counterpart (SMPLer-X requires task tokens to estimate body pose parameters, please see Supplementary Material Sec. C.1 for details), we have also trained a Hand4Whole [34], as the CNN-based foundation model (ResNet50 backbone for body and hand, ResNet18 for face with a total of 77M trainable parameters) to check if the conclusion still holds for CNN-based architectures. We highlight that similar to SMPLer-X, scaling up the training data for a CNN-based foundation model also shows a substantial performance boost.
>
> | Model | Sampling strategy | #Instance | GPU hours |MPE(mm)|
> | --- | --- | --- |--- | --- |
> | H4W-original | - | 0.65M | - | 115.59 |
> | H4W-32 | balanced | 4.5M | 230hr/epoch | 98.33 |
> | H4W-32 | concat | 5.6M | 290hr/epoch | 96.90 |
>
> ---
>
> > **Q2**: A few typos could be corrected.
>
> **A2**: Thank you for your suggestion, we will make the necessary amendments in the revised version.

---

> ### Author Response · Authors · 2023-08-28
> **Follow-up on Author Response**
>
> Dear Reviewer,
>
> We deeply appreciate the feedback you provided on our manuscript, which is critical to the further enhancement of the quality of our work. We hope that we have addressed your concerns in our previous response. Should there be any additional questions or areas you would like us to clarify further, please do not hesitate to let us know.
>
> Thank you very much for your attention.

---

> > ### Comment · Reviewer_6mpM · 2023-08-28
> >
> > Thanks for the clarifications. Most of my concerns are addressed. Combined with other reviewers' comments, I'd like to suggest including the additional experiments posted here in the final version and exploring more CNN-based models as well. Now, the strengths outweigh the weaknesses, therefore I keep my original rating.

---

### Official Review · Reviewer_nrrt · 2023-07-22
**Scaling Up Expressive Human Pose and Shape Estimation**

**Rating:** 7
**Confidence:** 4
**Clarity:** 1. L205, How does the claim that perf…

**Strengths:**

1. The investigated problem in this paper is intriguing and important.

2. The authors' efforts in data and model scaling are commendable. The presented approach surpasses previous work with significant performance improvements, highlighting the strength of their methodology.

3. The paper is well-written, and the authors have made their code openly available The accompanying document is also comprehensive, providing sufficient information.

**Additional Feedback:**

N/A

**Correctness:**

Overall the claims made in the paper are correct, and the evaluation methods and experiment design appropriate and performed correctly

**Documentation:**

there is sufficient detail in the document in the repo.

**Ethics:**

No.

**Limitations:**

The potential negative societal impact is well discussed in the paper

**Opportunities For Improvement:**

 In order to enhance the quality and comprehensiveness of the paper, it may be beneficial to include a video presentation or visual representation that showcases the qualitative results.

**Relation To Prior Work:**

The discussion on prior work is sufficient.

**Summary And Contributions:**

This paper contributes significantly to the development of EHPS by presenting a foundational model, addressing data scaling challenges, and exploring the use of vision transformers. Contribution includes:

1. Development of a foundational model by leveraging a large-scale backbone model and diverse datasets. The authors investigate the potential of vision transformers for model scaling in EHPS, examining the use of transformers and suggesting further improvements through fine-tuning

2.  To tackle the inconsistency among benchmarks, the authors optimize the training scheme and carefully select datasets.

---

> ### Author Response · Authors · 2023-08-22
> **Author Response (1/2)**
>
> We would like to express our sincere gratitude for your insightful comments.
>
> ---
>
> > **Q1**: It may be beneficial to include a video presentation or visual representation that showcases the qualitative results
>
> **A1**: Thank you for your suggestion. We have put several short clips with SMPL-X overlay in our [public repo](https://github.com/caizhongang/SMPLer-X), under the Gallery section. In addition, we provide qualitative results in Figure 4 and Supplementary Material Figure 12 to compare SMPLer-X with other strong methods under various challenges (diverse pose, complicated background, and severe occlusion/truncation). We will compile more visual representations to be released as a long demo video.
>
> ---
>
> > **Q2**: L205, How does the claim that performance is not strongly associated with the number of training instances, once the instance number exceeds approximately 30K, hold true for different network architectures and the number of parameters?
>
> **A2**: Thank you for raising the concern. We have compiled a [new figure](https://github.com/caizhongang/SMPLer-X/blob/main/assets/factor_data.pdf) that benchmarks two new base methods: Hand4Whole (CNN-based method) and SMPLer-X-S (SMPLer-X with ViT-S as the backbone). We highlight that training on 32 datasets individually and evaluating on five benchmarks are expensive to conduct (it takes 6400 GPU hours for Hand4Whole and 5100 GPU hours for SMPLer-X-S), hence, it is challenging to include more base methods within the short time constraint of the rebuttal period.
>
> Nevertheless, we observe that on the current 32 datasets, the general trend holds true across architectures and model sizes: as long as the number of training instances exceeds a certain number *n*, the dataset performance is no longer strongly associated with the number of training instances. Based on our observation, a rule of thumb estimation of *n* can be between 30K (the minimum estimation) to 100K (a more conservative estimation). We will revise our claim to add more discussions.
>
> ---
>
> > **Q3**: Meaning of “top 20, top 10, and SMPLer-X-L10”
>
> **A3**: Thank you for your suggestion. We will provide more details on this matter in the revised version. Specifically, we order all datasets in terms of MPE in the benchmark (Table 1), and “Top N” refers to the N datasets with the lowest MPE; foundation models are named “SMPLer-X-MN”, where M can be {S, B, L, H} that indicates the size of the ViT backbone, N indicates the number of datasets used in the training. For example, SMPLer-X-L10 means the foundation model with ViT-L as the backbone, trained with Top 10 datasets.

---

> > ### Author Response · Authors · 2023-08-22
> > **Author Response (2/2)**
> >
> > > **Q4**: The best performance is achieved when using all datasets. It appears that the claim in the abstract, stating that the authors optimized the training scheme and selected datasets leading to a significant leap, corresponds to the balanced sampling strategy. Is there any ablation study conducted to support this claim?
> >
> > **A4**: We performed the ablation study for the training scheme and sampling strategy separately.
> >
> > As for the training scheme, we investigated the effect of dataset selection. We selected the bottom 5 and bottom 10 datasets according to our single dataset benchmark rankings and trained the SMPLer-X-B model with the same number of instances as used in training with the top 5 and top 10 datasets.
> >
> > The table below shows that:
> > 1. Our training scheme is efficient. Selecting the top 5 or top 10 datasets according to the single dataset benchmark leads to a much better performance compared to selecting the bottom 5 or bottom 10 datasets.
> > 2. The foundation model can benefit from adding higher-ranked (i.e., Top 5/10) data into training, while lower-ranked data (i.e., Bottom 5/10) is not as effective in improving the model's performance.
> >
> > | Model | Dataset| #Instance | MPE(mm) |
> > | --- | --- | --- |--- |
> > | SMPLer-X-B | Top5 | 0.75M | 103.47 |
> > | SMPLer-X-B | Bottom5 | 0.75M | 115.61 |
> > | SMPLer-X-B | Top10 | 1.5M | 89.20 |
> > | SMPLer-X-B | Bottom10 | 1.5M |115.10 |
> >
> > As for the sampling strategy, we did the ablation study on three different strategies, including
> > 1. Balanced: we set all the datasets to have the same length as mentioned in the paper (L254-L257).
> > 2. Weighted: we set the dataset length according to the benchmark (Table 1) rankings. Specifically, we sort the datasets based on their rankings and then assign weights to each dataset. As a result of this weighting, the datasets are upsampled or downsampled so that the lengths of the datasets are adjusted to an arithmetic sequence. The length of the dataset with the highest ranking is 4 times that of the dataset with the lowest ranking, and the sum of the total lengths of all datasets is fixed.
> > 3. Concat: we simply concatenate all the datasets with their original length.
> >
> > The table below shows that the performance of the foundation model is not sensitive to the sampling strategy, while the balanced strategy is more intuitive, easy to implement, and efficient, the weighted strategy may have more potential with more effort in weight tuning.
> > | Model | Sampling strategy | #Instance | MPE(mm)|
> > | --- | --- | --- |--- |
> > | SMPLer-X-H32 | balanced | 4.5M | 63.08 |
> > | SMPLer-X-H32 | weighted | 4.5M | 62.12 |
> > | SMPLer-X-H32 | concat | 5.6M | 63.32 |

---

> ### Author Response · Authors · 2023-08-28
> **Follow-up on Author Response**
>
> Dear Reviewer,
>
> Thank you for your constructive feedback on our manuscript. Your comments were extremely valuable to the improvement of our work, and we hope that our previous response has adequately addressed your concerns. Please kindly advise us if any further questions or clarifications are needed, and we are committed to providing a prompt reply at your request.
>
> Once again, we would like to express our sincerest gratitude for your consideration.

---

### Official Review · Reviewer_er9Y · 2023-07-31
**SMPLer-X: Scaling Up Expressive Human Pose and Shape Estimation**

**Rating:** 7
**Confidence:** 4
**Correctness:** yes.
**Clarity:** yes.

**Strengths:**

To achieve a general and accurate EHPS, this paper explores more data combinations and model scaling methods. The proposed pretrained model is generalist foundation model across various scenarios and can be finetuned for any specific scenarios. The motivation for this paper is well justified, and implemented, and the overall organization of the paper flows well.

**Additional Feedback:**

Nothing yet.

**Documentation:**

There is sufficient detail on data collection and organization.

**Ethics:**

I do not suspect any ethical concerns with the submission.

**Limitations:**

The authors adequately describe the limitations of their work. 1) The comprehensive benchmark datasets is insufficient to represent the real-world distribution. 2) Finetuning harms the generalization capability of foundation models.

**Opportunities For Improvement:**

This paper provides insights into the strengths and limitations of current approaches, highlighting the potential of building generalist foundation model with both data and model scaling. However, 1) This paper requires more proofreading; 2) The description in the experimental section should be clearer.

**Relation To Prior Work:**

yes.

**Summary And Contributions:**

This paper explored both data and model scaling in building generalist benchmark on comprehensive EHPS datasets, and elevated our foundation model into a potent specialist across a multitude of benchmarks through data selection strategy.

---

> ### Author Response · Authors · 2023-08-22
> **Author Response**
>
> We are very grateful for your constructive comments.
>
> ---
>
> > **Q1**: This paper requires more proofreading
>
> **A1**: Thank you for your suggestion, we will run through the entire paper and correct any mistakes in the revised version.
>
> ----
>
> > **Q2**: The description in the experimental section should be clearer.
>
> **A2**: Thank you for your comments, we will revise the paper to make it clearer.
>
> - For dataset benchmark experiments:
>     - We will highlight that the preprocessing steps are included in Supplementary Material Sec. B, and datasets are not downsampled in these experiments.
>     - We will add hyperparameter settings (total batch size is 128, the learning rate is 1e-5, and the training takes place on 4 V100 GPUs for 5 epochs).
>
> - For foundation model experiments:
>     - We will specify the dataset used in each training in addition to the current “Top X” representation, and clearly define the naming pattern of our foundation models.
>     - We will highlight that the implementation and training details are included in Supplementary Material Sec. C
>     - We have also open-sourced our training configurations and codes in our [public repo](https://github.com/caizhongang/SMPLer-X) to disclose full details.
>
> - Moreover, we will make necessary re-arrangements of our chapters to clarify our experiment procedures.

---

> ### Author Response · Authors · 2023-08-28
> **Follow-up on Author Response**
>
> Dear Reviewer,
>
> We want to express our sincere gratitude for your insightful suggestions which are instrumental in enhancing the quality of our work. We hope that our proposed modifications would have addressed your concerns about the clarity of our presentation. We would really appreciate it if you could let us know if there are any further questions or aspects of the paper that require additional clarification.
>
> Thank you once again for your time and consideration.

---

### Author Response · Authors · 2023-08-22
**General Response**

We wish to convey our profound appreciation for the reviewers' acknowledgment of the strengths in our work:

- The problem is interesting with well-justified motivation (Reviewer **er9Y** and **nrrt**)
- Comprehensive investigation with deep insights and useful conclusions (Reviewer **6mpM** and **5yuA**)
- Significant performance improvements and provided pre-trained generalist models (Reviewer **nrrt**, **er9Y**, **6mpM**, **uVnR** and **5yuA**)

Moreover, we are indebted to the reviewers for their insightful comments. We address each reviewer’s concerns in separate comment threads.

---

> ### Comment · Area_Chair_i5Hk · 2023-08-29
> **Reviewers: please read authors' responses and share your thoughts and additional questions**
>
> Dear reviewers,
>
> This paper got a bit of mixed ratings, and we seek to achieve a consensus.
> We kindly request that you review the authors' rebuttal and indicate whether your concerns have been satisfactorily addressed.
>
> Reviewer 5yuA: We are keen to hear your perspective after considering the feedback from other reviewers and the authors' responses.
>
> Thank you for your valuable contributions to this process.

---

### Decision · Program_Chairs · 2023-09-22

**Decision:**

Accept (Poster)

**Comment:**

This paper got primarily positive ratings from reviewers, with scores of 7, 7, 7, 6, and 4. Reviewers who gave positive ratings appreciated the paper's exploration of the existing 32 datasets for Expressive Human Pose and Shape (EHPS) estimation. Reviewers viewed it as a valuable contribution, especially from the perspective of a foundation model, as it demonstrated robust performance in related benchmarks.

A reviewer who gave a negative rating expressed concerns about the paper's lack of presenting new data. In the authors' rebuttal, the authors defended their position, addressing that the paper still aligns with the NeurIPS Dataset and Benchmark track's goals.

Upon reviewing the D&B track's motivation, the AC concurred with the authors' perspective. The AC believes that the current paper effectively fulfills the track's objectives by offering a fresh viewpoint on consolidating and exploring the extensive array of existing datasets, thereby creating new value for the community, especially in the direction of building a foundation model for EHPS. The advantage of the proposed direction is well-supported by the experiments.

Considering the recognition of the paper's contributions by other reviewers, the AC concludes that the current version of the paper meets the bar of NeurIPS Datasets and Benchmarks Track. The authors are encouraged to carefully review the feedback provided by the reviewers and take steps to enhance the paper's quality for the camera-ready version.